

# The *ALF* (*A*lgorithms for *L*attice *F*ermions) project release 1.0 Documentation for the auxiliary field quantum Monte Carlo code

**Martin Bercx, Florian Goth, Johannes S. Hofmann and Fakher F. Assaad**

Institut für Theoretische Physik und Astrophysik, Universität Würzburg,
97074 Würzburg, Germany

alf@physik.uni-wuerzburg.de

## Abstract

The Algorithms for Lattice Fermions package provides a general code for the finite temperature auxiliary field quantum Monte Carlo algorithm. The code is engineered to be able to simulate any model that can be written in terms of sums of single-body operators, of squares of single-body operators and single-body operators coupled to an Ising field with given dynamics. We provide predefined types that allow the user to specify the model, the Bravais lattice as well as equal time and time displaced observables. The code supports an MPI implementation. Examples such as the Hubbard model on the honeycomb lattice and the Hubbard model on the square lattice coupled to a transverse Ising field are provided and discussed in the documentation. We furthermore discuss how to use the package to implement the Kondo lattice model and the $SU(N)$-Hubbard-Heisenberg model. One can download the code from our Git instance at `https://alf.physik.uni-wuerzburg.de` and sign in to file issues.



You are free to share and benefit from this documentation as long as this license is preserved and proper attribution to the authors is given. For details, see the ALF project homepage alf.physik.uni-wuerzburg.de. Contact address: alf@physik.uni-wuerzburg.de.

# 1 Introduction

## 1.1 Motivation

The auxiliary field quantum Monte Carlo (QMC) approach is the algorithm of choice to simulate thermodynamic properties of a variety of correlated electron systems in the solid state and beyond [1–8]. Apart from the physics of the canonical Hubbard model [9,10], the topics one can investigate in detail include correlation effects in the bulk and on surfaces of topological insulators [11,12], quantum phase transitions between Dirac fermions and insulators [13–17], deconfined quantum critical points [18,19], topologically ordered phases [19], heavy fermion systems [20,21], nematic [22] and magnetic [23] quantum phase transitions in metals, antiferromagnetism in metals [24], superconductivity in spin-orbit split bands [25], $SU(N)$ symmetric models [26,27], long-ranged Coulomb interactions in graphene systems [28,29], cold atomic gasses [30], low energy nuclear physics [31], entanglement entropies and spectra [32–36], etc. This ever growing list of topics is based on algorithmic progress and on recent symmetry related insights [37–41] enabling one to find negative sign problem free formulations of a number of model systems with very rich phase diagrams.

Auxiliary field methods can be formulated in very different ways. The fields define the configuration space $C$. They can stem from the Hubbard-Stratonovich (HS) [42] transformation required to decouple the many-body interacting term into a sum of non-interacting problems, or they can correspond to bosonic modes with predefined dynamics such as phonons or gauge fields. In all cases, the result is that the grand-canonical partition function takes the form,

$$Z = \text{Tr}\left(e^{-\beta \hat{\mathcal{H}}}\right) = \sum_C e^{-S(C)}, \tag{1}$$

where $S$ is the action of non-interacting fermions subject to a space-time fluctuating auxiliary field. The high-dimensional integration over the fields is carried out stochastically. In this formulation of many body quantum systems, there is no reason for the action to be a real number. Thereby $e^{-S(C)}$ cannot be interpreted as a weight. To circumvent this problem one can adopt re-weighting schemes and sample $|e^{-S(C)}|$. This invariably leads to the so called negative sign problem with associated exponential computational scaling in system size and inverse temperature [43,44]. The sign problem is formulation dependent, and as mentioned above there has been tremendous progress at identifying an ever growing class of negative sign problem free models covering a rich domain of collective emergent phenomena. For continuous fields, the stochastic integrations can be carried out with Langevin dynamics or hybrid methods [45]. However, for many problems one can get away with discrete fields [46]. In this case, Monte Carlo importance sampling will often be put to use [47]. We note that due to the non-locality of the fermion determinant, see below, cluster updates, such as in the loop or stochastic series expansion algorithms for quantum spin systems [48–50], are hard to formulate for this class of problems. The search for efficient updating schemes that enable to move quickly within the configuration space defines ongoing challenges.

Formulations do not differ only by the choice of the fields, continuous or discrete, and the sampling strategy, but also by the formulation of the action itself. For a given field configura-

tion, integrating out fermionic degrees of freedom generically leads to a fermionic determinant of dimension $\beta N$ where $\beta$ corresponds to the inverse temperature and $N$ to the volume of the system. Working with this determinant leads to the Hirsch-Fye approach [51] and its time complexity which quantifies the computational effort is given by $\mathcal{O}(\beta N)^3$. [1] The Hirsch-Fye algorithm is the method of choice for impurity problems, but has generically been outperformed by a class of so-called continuous-time quantum Monte Carlo approaches [52–54]. One key point of continuous-time methods is that they are action based and thereby allow to handle retarded interactions obtained when integrating out fermion or boson baths. In high dimensions and/or at low temperatures, the cubic scaling originating from the fermionic determinant is expensive. To circumvent this, the hybrid Monte-Carlo approach [5,55] expresses the fermionic determinant in terms of a Gaussian integral thereby introducing a new variable in the Monte Carlo integration. The resulting algorithm is the method of choice for lattice gauge theories in 3+1 dimensions and has been used to provide ab-inito estimates of light hadron masses starting from quantum chromo dynamics [56].

The algorithm implemented in the ALF project lies between the above two *extremes*. We will keep the fermionic determinant, but formulate the problem so as to work only with $N \times N$ matrices. This Blankenbecler, Scalapino, Sugar (BSS) algorithm scales linearly in imaginary time $\beta$, but remains cubic in the volume $N$. Furthermore, the algorithm can be formulated either in a projective manner [3, 4], adequate to obtain zero temperature properties in the canonical ensemble, or at finite temperatures in the grand-canonical ensemble [2].

The aim of the ALF project is to introduce a general formulation of the *finite* temperature auxiliary field QMC method with discrete fields so as to quickly be able to play with different model Hamiltonians at minimal programming cost. We have summarized the essential aspects of the auxiliary field QMC approach in this documentation, and refer the reader to Refs. [7,57] for complete reviews. We will show in all details how to implement a variety of models, run the code, and produce results for equal time and time displaced correlation functions. The program code is written in Fortran according to the 2003 standard and is able to natively utilize MPI for massively parallel runs on todays supercomputing systems.

The ALF package is not the first open source project aimed at providing simulation tools for correlated quantum matter. The most notable package is certainly the ALPS library [58]. It is actively maintained and features a whole set of algorithms for strongly correlated quantum lattice models including Monte Carlo, exact diagonalization, and density matrix renormalization group codes. It however does not include the auxiliary field QMC algorithm offered by the ALF package. Other projects include QUEST [59], TRIQS [60], w2dynamics [61] and iQist [62]. IQist, TRIQS and w2dynamics focus on approximate solutions via the CT-HYB [52] algorithm within the dynamical mean field approximation. The QUEST project implements the same algorithm as in the ALF project but is currently restricted to the Hubbard model and it does not allow to easily incorporate different Hamiltonians.

The ALF source code is placed under the GNU GPL license. The project is currently hosted on servers of the university of Würzburg where we have set up a GitLab instance (https://alf.physik.uni-wuerzburg.de) aimed at encouraging community outreach. Each potential user can sign in, receive space for his ALF related projects and share them with others. This site serves the GitLab issue tracker as well as a wiki so that members can collect information they consider useful for the project. We have set up an E-Mail address for reaching the core developers at alf@physik.uni-wuerzburg.de.

---

[1]Here we implicitly assume the absence of a negative sign problem.

## 1.2 Definition of the Hamiltonian

The first and most fundamental part of the project is to define a general Hamiltonian which can accommodate a large class of models. Our approach is to express the model as a sum of one-body terms, a sum of two-body terms each written as a perfect square of a one body term, as well as a one-body term coupled to an Ising field with dynamics to be specified by the user. The form of the interaction in terms of sums of perfect squares allows us to use generic forms of discrete approximations to the HS transformation [63,64]. Symmetry considerations are imperative to enhance the speed of the code. We therefore include a *color* index reflecting an underlying $SU(N)$ color symmetry as well as a flavor index reflecting the fact that after the HS transformation, the fermionic determinant is block diagonal in this index.

The class of solvable models includes Hamiltonians $\hat{\mathscr{H}}$ that have the following general form:

$$\hat{\mathscr{H}} = \hat{\mathscr{H}}_T + \hat{\mathscr{H}}_V + \hat{\mathscr{H}}_I + \hat{\mathscr{H}}_{0,I} , \quad \text{where} \tag{2}$$

$$\hat{\mathscr{H}}_T = \sum_{k=1}^{M_T} \sum_{\sigma=1}^{N_{\text{col}}} \sum_{s=1}^{N_{\text{fl}}} \sum_{x,y}^{N_{\text{dim}}} \hat{c}^\dagger_{x\sigma s} T^{(ks)}_{xy} \hat{c}_{y\sigma s} \equiv \sum_{k=1}^{M_T} \hat{T}^{(k)} , \tag{3}$$

$$\hat{\mathscr{H}}_V = \sum_{k=1}^{M_V} U_k \left\{ \sum_{\sigma=1}^{N_{\text{col}}} \sum_{s=1}^{N_{\text{fl}}} \left[ \left( \sum_{x,y}^{N_{\text{dim}}} \hat{c}^\dagger_{x\sigma s} V^{(ks)}_{xy} \hat{c}_{y\sigma s} \right) + \alpha_{ks} \right] \right\}^2 \equiv \sum_{k=1}^{M_V} U_k \left( \hat{V}^{(k)} \right)^2 , \tag{4}$$

$$\hat{\mathscr{H}}_I = \sum_{k=1}^{M_I} \hat{Z}_k \left( \sum_{\sigma=1}^{N_{\text{col}}} \sum_{s=1}^{N_{\text{fl}}} \sum_{x,y}^{N_{\text{dim}}} \hat{c}^\dagger_{x\sigma s} I^{(ks)}_{xy} \hat{c}_{y\sigma s} \right) \equiv \sum_{k=1}^{M_I} \hat{Z}_k \hat{I}^{(k)} . \tag{5}$$

The indices and symbols have the following meaning:

- The number of fermion *flavors* is set by $N_{\text{fl}}$. After the HS transformation, the action will be block diagonal in the flavor index.

- The number of fermion *colors* is set by $N_{\text{col}}$. The Hamiltonian is invariant under $SU(N_{\text{col}})$ rotations. [2]

- Both the color and the flavor index can describe the spin degree of freedom, the choice depending on the spin symmetry of the simulated model and the HS transformation. This point is illustrated in the examples, see Secs. 4.1 and 4.2.

- $N_{\text{dim}}$ is the total number of spacial vertices: $N_{\text{dim}} = N_{\text{unit cell}} N_{\text{orbital}}$, where $N_{\text{unit cell}}$ is the number of unit cells of the underlying Bravais lattice and $N_{\text{orbital}}$ is the number of (spacial) orbitals per unit cell.

- The indices $x$ and $y$ label lattice sites where $x, y = 1, \cdots, N_{\text{dim}}$.

- Therefore, the matrices $T^{(ks)}$, $V^{(ks)}$ and $I^{(ks)}$ are of dimension $N_{\text{dim}} \times N_{\text{dim}}$.

- The number of interaction terms is labelled by $M_V$ and $M_I$. $M_T > 1$ would allow for a checkerboard decomposition.

- $\hat{c}^\dagger_{y\sigma s}$ is a second quantized operator that creates an electron in a Wannier state centered around lattice site $y$, with color $\sigma$, and flavor index $s$. The operators satisfy the anti-commutation relations:

$$\left\{ \hat{c}^\dagger_{y\sigma s}, \hat{c}_{y'\sigma's'} \right\} = \delta_{y,y'} \delta_{s,s'} \delta_{\sigma,\sigma'}, \text{ and } \left\{ \hat{c}_{y\sigma s}, \hat{c}_{y'\sigma's'} \right\} = 0. \tag{6}$$

---

[2]Note that in the code $N_{\text{col}} \equiv$ `N_SUN`.

The Ising part of the general Hamiltonian (2) is $\hat{\mathscr{H}}_{0,I} + \hat{\mathscr{H}}_I$ and has the following properties:

- $\hat{Z}_k$ is an Ising spin operator which corresponds to the Pauli matrix $\hat{\sigma}_z$. It couples to a general one-body term.

- The dynamics of the Ising spins is given by $\hat{\mathscr{H}}_{0,I}$. This term is not specified here; it has to be specified by the user and becomes relevant when the Monte Carlo update probability is computed in the code (see Sec. 4.4 for an example).

Note that the matrices $T^{(ks)}$, $V^{(ks)}$ and $I^{(ks)}$ explicitly depend on the flavor index $s$ but not on the color index $\sigma$. The color index $\sigma$ only appears in the second quantized operators such that the Hamiltonian is manifestly $SU(N_{\text{col}})$ symmetric. We also require the matrices $T^{(ks)}$, $V^{(ks)}$ and $I^{(ks)}$ to be Hermitian.

As we will detail below, the definition of the above Hamiltonian allows to tackle several non-trivial models and phenomena. There are however a number of model Hamiltonians that cannot be simulated with ALF. Since we have opted for discrete fields, the electron-phonon interaction is not included. Furthermore, continuous HS transformations, that turn out to be extremely useful to include long-range Coulomb interactions [28, 65, 66], are not accessible in the present form of the package.[3] In many cases such as in $^3$He, three - or more body interactions should be included to capture relevant exchange mechanisms [67, 68]. These higher order processes are not captured in the ALF since it is *limited* to two-body interactions. The formulation of the Hamiltonian, has an explicit global $U(1)$ symmetry corresponding to particle number conservation. Hence using the ALF for a given model implies the existence of a canonical transformation where a particle number is conserved. Imaginary time dependent Hamiltonians, required to compute Renyi entropies and entanglement spectra [33, 35, 36] are not yet in the scope of ALF. Finally, one should also mention that auxiliary field QMC simulations are Hamiltonian based such that retarded interactions are not included in the ALF. For this set of problems, CT-INT type approaches are the method of choice [53, 54, 69]. The above *short comings* partially define a set of future directions that will be discussed in the concluding part of this documentation.

## 1.3 Outline

To use the code, a minimal understanding of the algorithm is necessary. In Sec. 2, we go very briefly through the steps required to formulate the many-body imaginary-time propagation in terms of a sum over HS and Ising fields of one-body imaginary-time propagators. The user has to provide this one-body imaginary-time propagator for a given configuration of HS and Ising fields. We equally discuss the Monte Carlo updates, the strategies for numerical stabilization of the code, as well as the Monte Carlo sampling.

Section 3 is devoted to the data structures that are needed to implement the model, as well as to the input and output file structure. The data structure includes an `Operator` type to optimally work with sparse Hermitian matrices, a `Lattice` type to define one- and two-dimensional Bravais lattices, and two `Observable` types to handle scalar observables (e.g. total energy) and equal time or time displaced two- point correlation functions (e.g. spin-spin correlations).

The Monte Carlo run and the data analysis are separated: the QMC run dumps the results of *bins* sequentially into files which are then analyzed by analysis programs. In Sec. 3.5, we provide a brief description of the analysis programs for our observable types. The analysis programs allow for omitting a given number of initial bins in order to account for warmup times. Also, a rebinning analysis is included to a posteriori take account of long autocorrelation times. Finally, Sec. 3.6 provides all the necessary details to compile and run the code.

---

[3]Note however that one can readily add short ranged interactions by including terms such as $(\hat{n}_i + \hat{n}_j - 2)^2$.

In Sec. 4, we give explicit examples on how to use the code for the Hubbard model on square and honeycomb lattices, for different choices of the Hubbard-Stratonovich transformation (see Secs. 4.1, 4.2 and 4.3) as well as for the Hubbard model on a square lattice coupled to a transverse Ising field (see Sec. 4.4 ). Our implementation is rather general such that a variety of other models can be simulated. In Sec. 5 we provide some information on how to simulate the Kondo lattice model as well as the $SU(N)$ symmetric Hubbard-Heisenberg model.

Finally, in Sec. 6 we list a number of features that are considered for future releases of the ALF program package.

## 2 Auxiliary Field Quantum Monte Carlo

### 2.1 Formulation of the method

Our aim is to compute observables for the general Hamiltonian (2) in thermodynamic equilibrium as described by the grand-canonical ensemble. We will show below how the grand-canonical partition function is rewritten as

$$Z = \text{Tr}\left(e^{-\beta \hat{\mathscr{H}}}\right) = \sum_C e^{-S(C)} + \mathscr{O}(\Delta\tau^2) \tag{7}$$

and define the space of configurations $C$. Note that the chemical potential term is already included in the definition of the one-body term $\hat{\mathscr{H}}_T$, see eq. (3), of the general Hamiltonian.

The outline of this section is as follows. First, we derive the detailed form of the partition function and outline the computation of observables (Sec. 2.1.1 - 2.1.3). Next, we present the present update strategy, namely local updates (Sec. 2.2). We equally discuss the measures we have implemented to make the code numerically stable (Sec. 2.3). Finally, we discuss the autocorrelations and associated time scales during the Monte Carlo sampling process (Sec. 2.4).

The essential ingredients of the auxiliary field quantum Monte Carlo implementation in the ALF package are the following:

- We will discretize the imaginary time propagation: $\beta = \Delta\tau L_{\text{Trotter}}$. Generically this introduces a systematic Trotter error of $\mathscr{O}(\Delta\tau)^2$ [70]. We note that there has been considerable effort at getting rid of the Trotter systematic error and to formulate a genuine continuous-time BSS algorithm [71]. To date, efforts in this direction are based on a CT-AUX type formulation [72,73] and face two issues. The first issue is that they are restricted to a class of models with Hubbard-type interactions

$$(\hat{n}_i - 1)^2 = (\hat{n}_i - 1)^4 \,, \tag{8}$$

such that the basic CT-AUX equation [74]

$$1 + \frac{U}{K}(\hat{n}_i - 1)^2 = \frac{1}{2}\sum_{s=\pm 1} e^{\alpha s(\hat{n}_i - 1)} \ \text{ with } \ \frac{U}{K} = \cosh(\alpha) - 1 \ \text{ and } \ K \in \mathbb{R} \tag{9}$$

holds. The second issue is that in the continuous-time approach it is hard to formulate a computationally efficient algorithm. Given this situation it turns out that the multi-grid method [75–77] is an efficient scheme to extrapolate to small imaginary-time steps so as to eliminate the Trotter systematic error if required.

- Having isolated the two-body term, we will use the discrete HS transformation [63,64]:

$$e^{\Delta\tau\lambda\hat{A}^2} = \sum_{l=\pm 1,\pm 2} \gamma(l)e^{\sqrt{\Delta\tau\lambda}\eta(l)\hat{A}} + \mathscr{O}(\Delta\tau^4) \,, \tag{10}$$

where the fields $\eta$ and $\gamma$ take the values:

$$\gamma(\pm 1) = 1 + \sqrt{6}/3, \quad \eta(\pm 1) = \pm\sqrt{2\left(3 - \sqrt{6}\right)}, \tag{11}$$

$$\gamma(\pm 2) = 1 - \sqrt{6}/3, \quad \eta(\pm 2) = \pm\sqrt{2\left(3 + \sqrt{6}\right)}.$$

Since the Trotter error is already of order $(\Delta\tau^2)$ per time slice, this transformation is next to exact.

- We will work in a basis for the Ising spins where $\hat{Z}_k$ is diagonal: $\hat{Z}_k|s_k\rangle = s_k|s_k\rangle$, where $s_k = \pm 1$.

- From the above it follows that the Monte Carlo configuration space $C$ is given by the combined spaces of Ising spin configurations and of HS discrete field configurations:

$$C = \left\{ s_{i,\tau}, l_{j,\tau} \text{ with } i = 1\cdots M_I, \ j = 1\cdots M_V, \ \tau = 1\cdots L_{\text{Trotter}} \right\}. \tag{12}$$

Here, the Ising spins take the values $s_{i,\tau} = \pm 1$ and the HS fields take the values $l_{j,\tau} = \pm 2, \pm 1$.

### 2.1.1 The partition function

With the above, the partition function of the model (2) can be written as follows.

$$
\begin{aligned}
Z &= \text{Tr}\left( e^{-\beta\hat{\mathcal{H}}} \right) \\
&= \text{Tr}\left[ e^{-\Delta\tau\hat{\mathcal{H}}_{0,I}} \prod_{k=1}^{M_V} e^{-\Delta\tau U_k\left(\hat{V}^{(k)}\right)^2} \prod_{k=1}^{M_I} e^{-\Delta\tau\hat{\sigma}_k\hat{I}^{(k)}} \prod_{k=1}^{M_T} e^{-\Delta\tau\hat{T}^{(k)}} \right]^{L_{\text{Trotter}}} + \mathcal{O}(\Delta\tau^2) \\
&= \sum_C \left( \prod_{k=1}^{M_V} \prod_{\tau=1}^{L_{\text{Trotter}}} \gamma_{k,\tau} \right) e^{-S_{0,I}(\{s_{i,\tau}\})} \times \\
&\quad \text{Tr}_{\text{F}}\left\{ \prod_{\tau=1}^{L_{\text{Trotter}}} \left[ \prod_{k=1}^{M_V} e^{\sqrt{-\Delta\tau U_k}\,\eta_{k,\tau}\hat{V}^{(k)}} \prod_{k=1}^{M_I} e^{-\Delta\tau s_{k,\tau}\hat{I}^{(k)}} \prod_{k=1}^{M_T} e^{-\Delta\tau\hat{T}^{(k)}} \right] \right\} + \mathcal{O}(\Delta\tau^2). \tag{13}
\end{aligned}
$$

In the above, the trace Tr runs over the Ising spins as well as over the fermionic degrees of freedom, and $\text{Tr}_{\text{F}}$ only over the fermionic Fock space. $S_{0,I}\left(\{s_{i,\tau}\}\right)$ is the action corresponding to the Ising Hamiltonian, and is only dependent on the Ising spins so that it can be pulled out of the fermionic trace. We have adopted the short hand notation $\eta_{k,\tau} = \eta(l_{k,\tau})$ and $\gamma_{k,\tau} = \gamma(l_{k,\tau})$. At this point, and since for a given configuration $C$ we are dealing with a free propagation, we can integrate out the fermions to obtain a determinant:

$$
\begin{aligned}
\text{Tr}_{\text{F}}&\left\{ \prod_{\tau=1}^{L_{\text{Trotter}}} \left[ \prod_{k=1}^{M_V} e^{\sqrt{-\Delta\tau U_k}\,\eta_{k,\tau}\hat{V}^{(k)}} \prod_{k=1}^{M_I} e^{-\Delta\tau s_{k,\tau}\hat{I}^{(k)}} \prod_{k=1}^{M_T} e^{-\Delta\tau\hat{T}^{(k)}} \right] \right\} = \\
&\prod_{s=1}^{N_{\text{fl}}} \left[ e^{\sum_{k=1}^{M_V}\sum_{\tau=1}^{L_{\text{Trotter}}} \sqrt{-\Delta\tau U_k}\,\alpha_{k,s}\eta_{k,\tau}} \right]^{N_{\text{col}}} \times \\
&\prod_{s=1}^{N_{\text{fl}}} \left[ \det\left( \mathbb{1} + \prod_{\tau=1}^{L_{\text{Trotter}}} \prod_{k=1}^{M_V} e^{\sqrt{-\Delta\tau U_k}\,\eta_{k,\tau}V^{(ks)}} \prod_{k=1}^{M_I} e^{-\Delta\tau s_{k,\tau}I^{(ks)}} \prod_{k=1}^{M_T} e^{-\Delta\tau T^{(ks)}} \right) \right]^{N_{\text{col}}}, \tag{14}
\end{aligned}
$$

where the matrices $\mathbf{T}^{(ks)}$, $\mathbf{V}^{(ks)}$, and $\mathbf{I}^{(ks)}$ define the Hamiltonian [Eq. (2) - (5)]. All in all, the partition function is given by:

$$
Z = \sum_C e^{-S_{0,I}(\{s_{i,\tau}\})} \left( \prod_{k=1}^{M_V} \prod_{\tau=1}^{L_{\text{Trotter}}} \gamma_{k,\tau} \right) e^{N_{\text{col}} \sum_{s=1}^{N_{\text{fl}}} \sum_{k=1}^{M_V} \sum_{\tau=1}^{L_{\text{Trotter}}} \sqrt{-\Delta\tau U_k} \alpha_{k,s} \eta_{k,\tau}} \times
$$

$$
\prod_{s=1}^{N_{\text{fl}}} \left[ \det\left( \mathbb{1} + \prod_{\tau=1}^{L_{\text{Trotter}}} \prod_{k=1}^{M_V} e^{\sqrt{-\Delta\tau U_k}\eta_{k,\tau}\mathbf{V}^{(ks)}} \prod_{k=1}^{M_I} e^{-\Delta\tau s_{k,\tau}\mathbf{I}^{(ks)}} \prod_{k=1}^{M_T} e^{-\Delta\tau \mathbf{T}^{(ks)}} \right) \right]^{N_{\text{col}}} + \mathcal{O}(\Delta\tau^2)
$$

$$
\equiv \sum_C e^{-S(C)} + \mathcal{O}(\Delta\tau^2). \tag{15}
$$

In the above, one notices that the weight factorizes in the flavor index. The color index raises the determinant to the power $N_{\text{col}}$. This corresponds to an explicit $SU(N_{\text{col}})$ symmetry for each configuration. This symmetry is manifest in the fact that the single particle Green functions are color independent, again for each given configuration $C$.

### 2.1.2 Observables

In the auxiliary field QMC approach, the single-particle Green function plays a crucial role. It determines the Monte Carlo dynamics and is used to compute observables:

$$
\langle \hat{O} \rangle = \frac{\text{Tr}\left[ e^{-\beta\hat{H}}\hat{O} \right]}{\text{Tr}\left[ e^{-\beta\hat{H}} \right]} = \sum_C P(C) \langle\langle \hat{O} \rangle\rangle_{(C)}, \text{ with } P(C) = \frac{e^{-S(C)}}{\sum_C e^{-S(C)}} . \tag{16}
$$

$\langle\langle \hat{O} \rangle\rangle_{(C)}$ corresponds to the expectation value of $\hat{O}$ for a given configuration $C$. For a given configuration $C$ one can use Wick's theorem to compute $\langle\langle \hat{O} \rangle\rangle_{(C)}$ from the knowledge of the single-particle Green function:

$$
G(x,\sigma,s,\tau|x',\sigma',s',\tau') = \langle\langle \mathcal{T}\hat{c}_{x\sigma s}(\tau)\hat{c}^\dagger_{x'\sigma's'}(\tau') \rangle\rangle_C , \tag{17}
$$

where $\mathcal{T}$ corresponds to the imaginary-time ordering operator. The corresponding equal time quantity reads:

$$
G(x,\sigma,s,\tau|x',\sigma',s',\tau) = \langle\langle \hat{c}_{x\sigma s}(\tau)\hat{c}^\dagger_{x'\sigma's'}(\tau) \rangle\rangle_C. \tag{18}
$$

Since for a given HS field translation invariance in imaginary-time is broken, the Green function has an explicit $\tau$ and $\tau'$ dependence. On the other hand it is diagonal in the flavor index, and independent on the color index. The latter reflects the explicit $SU(N)$ color symmetry present at the level of individual HS configurations. As an example, one can show that the equal time Green function at $\tau = 0$ reads [7]:

$$
G(x,\sigma,s,0|x',\sigma,s,0) = \left( \mathbb{1} + \prod_{\tau=1}^{L_{\text{Trotter}}} \mathbf{B}_\tau^{(s)} \right)^{-1}_{x,x'} \tag{19}
$$

with

$$
\mathbf{B}_\tau^{(s)} = \prod_{k=1}^{M_T} e^{-\Delta\tau \mathbf{T}^{(ks)}} \prod_{k=1}^{M_V} e^{\sqrt{-\Delta\tau U_k}\eta_{k,\tau}\mathbf{V}^{(ks)}} \prod_{k=1}^{M_I} e^{-\Delta\tau s_{k,\tau}\mathbf{I}^{(ks)}}. \tag{20}
$$

To compute equal time as well as time displaced observables, one can make use of Wick's

theorem. A convenient formulation of this theorem for QMC simulations reads:

$$
\langle\langle \mathscr{T} c^\dagger_{\underline{x}_1}(\tau_1) c_{\underline{x}'_1}(\tau'_1) \cdots c^\dagger_{\underline{x}_n}(\tau_n) c_{\underline{x}'_n}(\tau'_n) \rangle\rangle_C =
$$

$$
\det \begin{bmatrix}
\langle\langle \mathscr{T} c^\dagger_{\underline{x}_1}(\tau_1) c_{\underline{x}'_1}(\tau'_1) \rangle\rangle_C & \langle\langle \mathscr{T} c^\dagger_{\underline{x}_1}(\tau_1) c_{\underline{x}'_2}(\tau'_2) \rangle\rangle_C & \cdots & \langle\langle \mathscr{T} c^\dagger_{\underline{x}_1}(\tau_1) c_{\underline{x}'_n}(\tau'_n) \rangle\rangle_C \\
\langle\langle \mathscr{T} c^\dagger_{\underline{x}_2}(\tau_2) c_{\underline{x}'_1}(\tau'_1) \rangle\rangle_C & \langle\langle \mathscr{T} c^\dagger_{\underline{x}_2}(\tau_2) c_{\underline{x}'_2}(\tau'_2) \rangle\rangle_C & \cdots & \langle\langle \mathscr{T} c^\dagger_{\underline{x}_2}(\tau_2) c_{\underline{x}'_n}(\tau'_n) \rangle\rangle_C \\
\vdots & \vdots & \ddots & \vdots \\
\langle\langle \mathscr{T} c^\dagger_{\underline{x}_n}(\tau_n) c_{\underline{x}'_1}(\tau'_1) \rangle\rangle_C & \langle\langle \mathscr{T} c^\dagger_{\underline{x}_n}(\tau_n) c_{\underline{x}'_2}(\tau'_2) \rangle\rangle_C & \cdots & \langle\langle \mathscr{T} c^\dagger_{\underline{x}_n}(\tau_n) c_{\underline{x}'_n}(\tau'_n) \rangle\rangle_C
\end{bmatrix}. \quad (21)
$$

Here, we have defined the super-index $\underline{x} = \{x, \sigma, s\}$. In the subroutines `Obser` and `ObserT` of the module `Hamiltonian_Examples.f90` (see Sec. 3.2) the user is provided with the equal time and time displaced correlation function. Using the above formulation of Wick's theorem, arbitrary correlation functions can be computed. We note however, that the program is limited to the calculation of observables that contain only two different imaginary times.

### 2.1.3 Reweighting and the sign problem

In general, the action $S(C)$ will be complex, thereby inhibiting a direct Monte Carlo sampling of $P(C)$. This leads to the infamous sign problem. The sign problem is formulation dependent and as noted above, much progress has been made at understanding the class of models that can be formulated without encountering this problem [37–40]. When the average sign is not too small, we can nevertheless compute observables within a reweighting scheme. Here we adopt the following scheme. First note that the partition function is real such that:

$$
Z = \sum_C e^{-S(C)} = \sum_C \overline{e^{-S(C)}} = \sum_C \Re\left[e^{-S(C)}\right]. \quad (22)
$$

Thereby[4], and with the definition

$$
\text{sign}\,(C) = \frac{\Re\left[e^{-S(C)}\right]}{\left|\Re\left[e^{-S(C)}\right]\right|}, \quad (23)
$$

the computation of the observable [Eq. (16)] is re-expressed as follows:

$$
\begin{aligned}
\langle \hat{O} \rangle &= \frac{\sum_C e^{-S(C)} \langle\langle \hat{O} \rangle\rangle_{(C)}}{\sum_C e^{-S(C)}} \\
&= \frac{\sum_C \Re\left[e^{-S(C)}\right] \frac{e^{-S(C)}}{\Re\left[e^{-S(C)}\right]} \langle\langle \hat{O} \rangle\rangle_{(C)}}{\sum_C \Re\left[e^{-S(C)}\right]} \\
&= \frac{\left\{\sum_C \left|\Re\left[e^{-S(C)}\right]\right| \, \text{sign}\,(C) \frac{e^{-S(C)}}{\Re\left[e^{-S(C)}\right]} \langle\langle \hat{O} \rangle\rangle_{(C)}\right\} / \sum_C \left|\Re\left[e^{-S(C)}\right]\right|}{\left\{\sum_C \left|\Re\left[e^{-S(C)}\right]\right| \, \text{sign}\,(C)\right\} / \sum_C \left|\Re\left[e^{-S(C)}\right]\right|} \\
&= \frac{\left\langle \text{sign}\, \frac{e^{-S}}{\Re[e^{-S}]} \langle\langle \hat{O} \rangle\rangle \right\rangle_{\overline{P}}}{\langle \text{sign} \rangle_{\overline{P}}}.
\end{aligned} \quad (24)
$$

The average sign is

$$
\langle \text{sign} \rangle_{\overline{P}} = \frac{\sum_C \left|\Re\left[e^{-S(C)}\right]\right| \, \text{sign}\,(C)}{\sum_C \left|\Re\left[e^{-S(C)}\right]\right|}, \quad (25)
$$

---

[4]The attentive reader will have noticed that for arbitrary Trotter decompositions, the imaginary time propagator is not necessarily Hermitian. Thereby, the above equation is correct only up to corrections stemming from the controlled Trotter systematic error.

and we have $\langle \text{sign} \rangle_{\overline{P}} \in \mathbb{R}$ per definition. The Monte Carlo simulation samples the probability distribution

$$\overline{P}(C) = \frac{\left| \Re\left[ e^{-S(C)} \right] \right|}{\sum_C \left| \Re\left[ e^{-S(C)} \right] \right|} \, . \tag{26}$$

such that the nominator and denominator of Eq. (24) can be computed.

The negative sign problem is an issue since the average sign is a ratio of two partition functions such that one can argue that [43]

$$\langle \text{sign} \rangle_{\overline{P}} \propto e^{-\Delta N \beta}. \tag{27}$$

$\Delta$ is intensive positive quantity and $N\beta$ denotes the Euclidean volume. In a Monte Carlo simulation, the error scales as $1/\sqrt{T_{\text{CPU}}}$ where $T_{\text{CPU}}$ corresponds to the computational time. Since the error on the average sign has to be much smaller than the average sign itself, one sees that:

$$T_{\text{CPU}} \gg e^{2\Delta N \beta}. \tag{28}$$

Two comments are in order. First, the presence of a sign problem invariably leads to an exponential increase of CPU time as a function of the Euclidean volume. And second, $\Delta$ is formulation dependent. For instance, at finite doping, the SU(2) invariant formulation of the Hubbard model presented in Sec. 4.1 has a much more severe sign problem than the formulation presented in Sec. 4.2 where the HS field couples to the z-component of the magnetization. Typically one can work with average signs down to $\langle \text{sign} \rangle_{\overline{P}} \simeq 0.1$.

## 2.2 Updating schemes

The program allows for different types of updating schemes. Given a configuration $C$ we propose a new one, $C'$, with probability $T_0(C \to C')$ and accept it according to the Metropolis-Hastings acceptance-rejection probability,

$$P(C \to C') = \min\left( 1, \frac{T_0(C' \to C) W(C')}{T_0(C \to C') W(C)} \right), \tag{29}$$

so as to guarantee the stationarity condition. Here, $W(C) = \left| \Re\left[ e^{-S(C)} \right] \right|$.

| Variable | Type | Description |
|---|---|---|
| `Propose_S0` | Logical | If true, proposes local moves according to the probability $e^{-S_{0,I}}$. |

Table 1: Variable required to control the updating scheme.

### 2.2.1 The default: sequential single spin flips

The default updating scheme is a sequential single spin flip algorithm. Consider the Ising spin $s_{i,\tau}$. We will flip it with probability one such that for this local move the proposal matrix is symmetric. If we are considering the Hubbard-Stratonovich field $l_{i,\tau}$ we will propose with probability 1/3 one of the other three possible fields. Again, for this local move, the proposal matrix is symmetric. Hence in both cases we will accept or reject the move according to

$$P(C \to C') = \min\left( 1, \frac{W(C')}{W(C)} \right). \tag{30}$$

It is worth noting that this type of sequential spin flip updating does not satisfy detailed balance but the more fundamental stationarity condition [47].

### 2.2.2 Sampling of $e^{-S_{0,I}}$

Consider an Ising spin at space-time $i, \tau$ and the configuration $C$. Flipping this spin will generate the configuration $C'$ and we will propose the move according to

$$T_0(C \rightarrow C') = \frac{e^{-S_{0,I}(C')}}{e^{-S_{0,I}(C')} + e^{-S_{0,I}(C)}} = 1 - \frac{1}{1 + e^{-S_{0,I}(C')}/e^{-S_{0,I}(C)}}. \tag{31}$$

Note that the function S0 in the `Hamitonian_example.f90` module computes precisely the ratio $e^{-S_{0,I}(C')}/e^{-S_{0,I}(C)}$ so that $T_0(C \rightarrow C')$ does not require any further programming. Thereby one will accept the proposed move with the probability:

$$P(C \rightarrow C') = \min\left(1, \frac{e^{-S_{0,I}(C)}W(C')}{e^{-S_{0,I}(C')}W(C)}\right). \tag{32}$$

With Eq. 15 one sees that the bare action $S_{0,I}(C)$ determining the dynamics of the Ising spin in the absence of coupling to the fermions does not enter the Metropolis acceptance-rejection step. This sampling scheme is used if the logical variable `Propose_S0` is set to `true`.

### 2.3 Stabilization - a peculiarity of the BSS algorithm

From (15) it can be seen that for the calculation of the Monte Carlo weight and for the observables a long product of matrix exponentials has to be formed. On top of that we need to be able to extract the single-particle Green function for a given flavor index at say time slice $\tau = 0$. As mentioned above in Eq. (19), this quantity is given by:

$$G = \left(\mathbb{1} + \prod_{\tau=1}^{L_{\text{Trotter}}} B_\tau\right)^{-1}. \tag{33}$$

To boil this down to more familiar terms from linear algebra we remark that we can recast this problem as the task to find the solution of the linear system

$$(\mathbb{1} + \prod_\tau B_\tau)x = b. \tag{34}$$

The $B_\tau \in \mathbb{C}^{n \times n}$ depend on the lattice size as well as other physical parameters that can be chosen such that a matrix norm of $B_\tau$ can be unbound in size. From standard perturbation theory for linear systems it is known that the computed solution $\tilde{x}$ would contain a relative error of

$$\frac{|\tilde{x} - x|}{|x|} = \mathcal{O}\left(\epsilon \kappa_p \left(\mathbb{1} + \prod_\tau B_\tau\right)\right). \tag{35}$$

Here $\epsilon$ denotes the machine precision, which is $2^{-53}$ for IEEE double precision numbers, and $\kappa_p(M)$ is the condition number of the matrix $M$ with respect to the matrix $p$-norm. The important property that makes straight-forward inversion so badly suited stems from the fact that $\prod_\tau B_\tau$ contains exponentially large and small scales as can be seen in Eq. (15). Thereby, as a function of increasing inverse temperature, the condition number will grow exponentially so that the computed solution $\tilde{x}$ will often contain no correct digits at all. To circumvent this, more sophisticated methods have to be employed. We will first of all assume that the multiplication of `NWrap` $B$ matrices has an acceptable condition number. Assuming for simplicity that $L_{\text{Trotter}}$ is an integer multiple of `NWrap`, we can write:

$$G = \left(\mathbb{1} + \prod_{i=0}^{\frac{L_{\text{Trotter}}}{\text{NWrap}}-1} \underbrace{\prod_{\tau=1}^{\text{NWrap}} B_{i \cdot \text{NWrap}+\tau}}_{\equiv \mathscr{B}_i}\right)^{-1}. \tag{36}$$

Within the auxiliary field QMC implementation of the ALF project, we are by default employing the strategy of forming a product of QR-decompositions which was proven to be weakly backwards stable in Ref. [78]. The key idea is to efficiently separate the scales of a matrix from the orthogonal part of a matrix. This can be achieved using a QR decomposition of a matrix $A$ in the form $A_i = Q_i R_i$. The matrix $Q_i$ is unitary and hence in the usual 2-norm it holds that $\kappa_2(Q_i) = 1$. To get a handle on the condition number of $R_i$ we will form the diagonal matrix

$$(D_i)_{n,n} = |(R_i)_{n,n}|, \tag{37}$$

and set $\tilde{R}_i = D_i^{-1} R_i$ This gives the decomposition

$$A_i = Q_i D_i \tilde{R}_i. \tag{38}$$

$D_i$ now contains the row norms of the original $R_i$ matrix and hence attempts to separate off the total scales of the problem from $R_i$. This is similar in spirit to the so-called matrix equilibration which tries to improve the condition number of a matrix by suitably chosen column and row scalings. Due to a theorem by van der Sluis [79] we know that the choice in Eq. (37) is almost optimal among all diagonal matrices $D$ from the space of diagonal matrices $\mathscr{D}$ in the sense that

$$\kappa_p((D_i)^{-1} R_i) \le n^{1/p} \min_{D \in \mathscr{D}} \kappa_p(D^{-1} R_i).$$

Now, given an initial decomposition of $A_{j-1} = \prod_i \mathscr{B}_i = Q_{j-1} D_{j-1} T_{j-1}$ an update $\mathscr{B}_j A_{j-1}$ is formed in the following three steps:

1. Form $M_j = (\mathscr{B}_j Q_{j-1}) D_{j-1}$. Note the parentheses.

2. Do a QR decomposition of $M_j = Q_j D_j R_j$. This gives the final $Q_j$ and $D_j$.

3. Form the updated $T$ matrices $T_j = R_j T_{j-1}$.

While this might seem like quite an effort that has to be performed for every multiplication it has to be noted that even with this stabilization scheme the algorithm preserves the time complexity class of $\mathcal{O}(\beta N^3)$ expressed in the physical parameters inverse temperature $\beta$ and lattice size $N$. While there is no analytical expression for the dependence of the stability on the physical parameters our experience has been that for a given number of stabilization steps along the imaginary time axis [in the notation of Eq. (36) this number is $L_{\text{Trotter}}/\texttt{NWrap}$], the precision will be largely invariant of the system size $N$, whereas with increasing inverse temperature $\beta$ the number of stabilization steps often has to be increased to maintain a given precision. The effectiveness of the stabilization *has* to be judged for every simulation from the output file `info` (Sec. 3.3.3). For most simulations there are two values to look out for:

- `Precision Green`

- `Precision Phase`

The Green function as well as the average phase are usually numbers with a magnitude of $\mathcal{O}(1)$. For that reason we recommend that `NWrap` is chosen such that the mean precision is of the order of $10^{-8}$ or better. We have included typical values of `Precision Phase` and of the mean and the maximal values of `Precision Green` in the discussion of example simulations, see Sec. 4.1.3 and Sec. 4.2.3.

## 2.4 Monte Carlo sampling

Error estimates in Monte Carlo simulations can be delicate and are based on the central limit theorem [80]. This theorem requires independent measurements and a finite variance. In this subsection we will give examples of the issues that a user will have to look out for while using a Monte Carlo code. Those effects are part of the common lore of the field and we can only touch on them briefly in this text. For a deeper understanding of the inherent issues of Markov chain Monte Carlo methods we refer the reader to the pedagogical introduction in chapter 1.3.5 of Krauth [81], the overview article of Sokal [47], the more specialized literature by Geyer [82] and chapter 6.3 of Neal [83].

In general, one distinguishes local from global updates. As the name suggest, the local update corresponds to a small change of the configuration, e.g. a single spin flip of one of the $L_{\text{Trotter}}(M_I + M_V)$ field entries (see Sec. 2.2), whereas a global update changes a significant part of the configuration. The default update scheme of the implementation at hand are local updates such that a minimum amount of moves is required to generate a independent configuration. The associated time scale is called the autocorrelation time, $T_{\text{auto}}$, and is generically dependent upon the choice of the observables.

Our unit of *sweeps* is defined such that each field is visited twice in a sequential propagation from $\tau = 0$ to $\tau = L_{\text{Trotter}}$ and back. A single sweep will generically not suffice to produce an independent configuration. In fact, the autocorrelation time $T_{\text{auto}}$ characterizes the required time scale to generate an independent values of $\langle\langle \hat{O} \rangle\rangle_C$ for the observable $O$. This has several consequences for the Monte Carlo simulation:

- First of all, we start from a randomly chosen field configuration such that one has to invest *at least* one, but generically much more, $T_{\text{auto}}$ to generate relevant, equilibrated configurations before reliable measurements are possible. This phase of the simulation is known as the warm-up or burn-in phase. In order to keep the code as flexible as possible (different simulations might have different autocorrelation times), measurements are taken from the very beginning. Instead, we provide the parameter `n_skip` for the analysis to ignore the first `n_skip` bins.

- Secondly, our implementation averages over a given amount of measurements set by the variable `NSWEEPS` before storing the results, known as one bin, on the disk. A bin corresponds to `NSWEEPS` sweeps. The error analysis requires statistically independent bins to generate reliable confidence estimates. If bins are to small (averaged over a period shorter then $T_{\text{auto}}$), the error bars are then typically underestimated. Most of the time, the autocorrelation time is unknown before the simulation is started. Sometimes the used compute cluster does not allow single runs long enough to generate appropriately sized bins. Therefore, we provide the `N_rebin` parameter that specifies how many bins are combined into a new bin during the error analysis. In general, one should check that a further increase of the bin size does not change the error estimate (For an explicit example, the reader is referred to Sec. 2.4.2 and the appendix of Ref. [57]).

  The `N_rebin` variable can be used to control a second issue. The distribution of the Monte Carlo estimates $\langle\langle \hat{O} \rangle\rangle_C$ is unknown. The result in the form (mean ± error) assumes a Gaussian distribution. Every original distribution with a finite variance turns into a Gaussian one, once it is folded often enough (central limit theorem). Due to the internal averaging (folding) within one bin, many observables are already quite Gaussian. Otherwise one can increase `N_rebin` further, even if the bins are already independent [84].

- The third issue concerns time displaced correlation functions. Even if the configurations are independent, the fields within the configuration are still correlated. Hence, the data

for $S_{\alpha,\beta}(\vec{k},\tau)$ (see Sec. 3.2; Eqn. 65) and $S_{\alpha,\beta}(\vec{k},\tau+\Delta\tau)$ are also correlated. Setting the switch N_Cov=1 triggers the calculation of the covariance matrix in addition to the usual error analysis. The covariance is defined by

$$COV_{\tau\tau'} = \frac{1}{N_{\text{Bin}}}\left\langle\left(S_{\alpha,\beta}(\vec{k},\tau)-\langle S_{\alpha,\beta}(\vec{k},\tau)\rangle\right)\left(S_{\alpha,\beta}(\vec{k},\tau')-\langle S_{\alpha,\beta}(\vec{k},\tau')\rangle\right)\right\rangle. \tag{39}$$

An example where this information is necessary is the calculation of mass gaps extracted by fitting the tail of the time displaced correlation function. Omitting the covariance matrix will underestimate the error.

### 2.4.1 The Jackknife resampling method

For each observable $\hat{A},\hat{B},\hat{C}\cdots$ the Monte Carlo program computes a data set of $N_{\text{Bin}}$ (ideally) independent values where for each observable the measurements belong to the same statistical distribution. In the general case, we would like to evaluate a function of expectation values, $f(\langle\hat{A}\rangle,\langle\hat{B}\rangle,\langle\hat{C}\rangle\cdots)$ – see for example the expression (24) for the observable including reweighting – and are interested in the statistical estimates of its mean value and the standard error of the mean. A numerical method for the statistical analysis of a given function $f$ which properly handles error propagation and correlations among the observables is the Jackknife method, which is, like the related Bootstrap method, a resampling scheme [85]. Here we briefly review the *delete-1 Jackknife* scheme which is based on the idea to generate $N_{\text{bin}}$ new data sets of size $N_{\text{bin}}-1$ by consecutively removing one data value from the original set. By $A_{(i)}$ we denote the arithmetic mean for the observable $\hat{A}$, without the $i$-th data value $A_i$, namely

$$A_{(i)} \equiv \frac{1}{N_{\text{Bin}}-1}\sum_{k=1,k\neq i}^{N_{\text{Bin}}}A_k. \tag{40}$$

As the corresponding quantity for the function $f(\langle\hat{A}\rangle,\langle\hat{B}\rangle,\langle\hat{C}\rangle\cdots)$, we define

$$f_{(i)}(\langle\hat{A}\rangle,\langle\hat{B}\rangle,\langle\hat{C}\rangle\cdots) \equiv f(A_{(i)},B_{(i)},C_{(i)}\cdots). \tag{41}$$

Following the convention in the literature, we will denote the final Jackknife estimate of the mean by $f_{(\cdot)}$ and its standard error by $\Delta f$. The Jackknife mean is given by

$$f_{(\cdot)}(\langle\hat{A}\rangle,\langle\hat{B}\rangle,\langle\hat{C}\rangle\cdots) = \frac{1}{N_{\text{Bin}}}\sum_{i=1}^{N_{\text{Bin}}}f_{(i)}(\langle\hat{A}\rangle,\langle\hat{B}\rangle,\langle\hat{C}\rangle\cdots), \tag{42}$$

and the standard error, including bias correction, is given by

$$(\Delta f)^2 = \frac{N_{\text{Bin}}-1}{N_{\text{Bin}}}\sum_{i=1}^{N_{\text{Bin}}}\left[f_{(i)}(\langle\hat{A}\rangle,\langle\hat{B}\rangle,\langle\hat{C}\rangle\cdots)-f_{(\cdot)}(\langle\hat{A}\rangle,\langle\hat{B}\rangle,\langle\hat{C}\rangle\cdots)\right]^2. \tag{43}$$

In case of $f=\langle\hat{A}\rangle$, the results (42) and (43) reduce to the plain sample average and the standard, bias corrected, estimate of the error.

### 2.4.2 An explicit example of error estimation

In the following we use one of our examples, the Hubbard model on a square lattice in the $M_z$ Hubbard-Stratonovich decoupling (see Sec. 4.2), to show explicitly how to estimate errors. We will equally show that the autocorrelation time is dependent upon the choice of the observable. In fact, different observables within the same run can have different autocorrelation times and

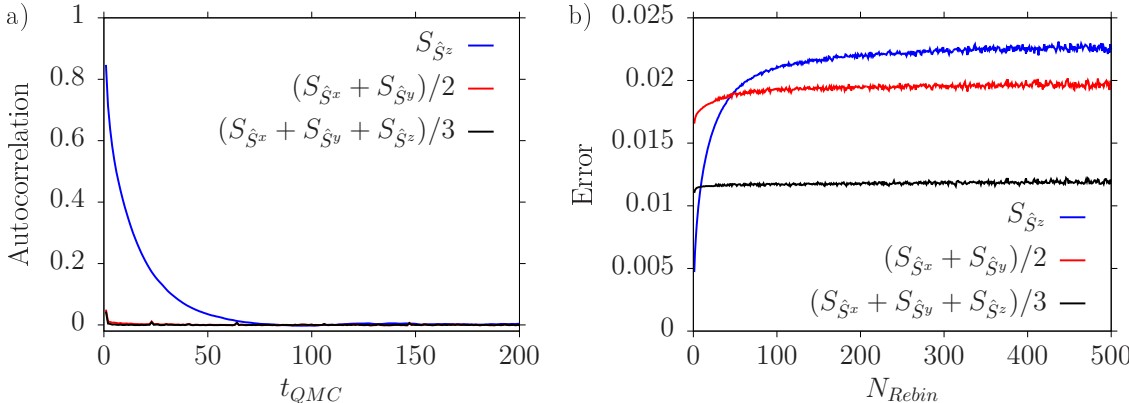

Figure 1: The autocorrelation function $Auto_{\hat{O}}(t_{\text{QMC}})$ (a) and the scaling of the error with effective bin size (b) of three equal time spin-spin correlation functions $\hat{O}$ of the Hubbard model in the $M_z$ decoupling (see Sec. 4.2). Simulations were done on a $6 \times 6$ square lattice, with $U/t = 4$ and $\beta t = 6$. The original bin contained only one sweep and we calculated around one million bins on a single core. The different autocorrelation times for the $xy$-plane compared to the $z$-direction can be detected from the decay rate of the autocorrelation function (a) and from the point where saturation of the error sets in (b), which defines the required effective bin size for independent measurements. Apparently and as argued in the text, the improved estimator $(S_{\hat{S}^x} + S_{\hat{S}^y} + S_{\hat{S}^z})/3$ has the smallest autocorrelation time.

of course, this time scale depends on the parameter choice. Hence, the user has to check autocorrelations of individual observables for each simulation! Typical regions of the phase diagram that require special attention are critical points where length scales diverge.

To determine the autocorrelation time, we calculate the correlation function

$$Auto_{\hat{O}}(t_{\text{QMC}}) = \sum_{i=0}^{N_{\text{Bin}}-t_{\text{QMC}}} \frac{\left(O_i - \langle \hat{O} \rangle\right)\left(O_{i+t_{\text{QMC}}} - \langle \hat{O} \rangle\right)}{\left(O_i - \langle \hat{O} \rangle\right)\left(O_i - \langle \hat{O} \rangle\right)}, \tag{44}$$

where $O_i$ refers to the Monte Carlo estimate of the observable $\hat{O}$ in the $i^{\text{th}}$ bin. This function typically shows an exponential decay and the decay rate defines the autocorrelation time. Figure 1 (a) shows the autocorrelation functions $Auto_{\hat{O}}(t_{\text{QMC}})$ for three spin-spin-correlation functions [Eq. (65)] at momentum $\vec{k} = (\pi, \pi)$ and at $\tau = 0$: $\hat{O} = S_{\hat{S}^z}$ for the $z$ spin direction, $\hat{O} = (S_{\hat{S}^x} + S_{\hat{S}^y})/2$ for the $xy$ plane, and $\hat{O} = (S_{\hat{S}^x} + S_{\hat{S}^y} + S_{\hat{S}^z})/3$ for the total spin. The Hubbard model has a $SU(2)$ spin symmetry. However, we chose a HS field which couples to the $z$-component of the magnetization, $M_z$, such that each configuration breaks this symmetry. Of course, after Monte Carlo averaging one expects restoration of the symmetry. The model, on bipartite lattices, shows spontaneous spin-symmetry breaking at $T = 0$ and in the thermodynamic limit. At finite temperatures, and within the so-called renormalized classical regime, quantum antiferromagnets have a length scale that diverges exponentially with decreasing temperatures [86]. The parameter set chosen for Fig. 1 is non-trivial in the sense that it places the Hubbard model in this renormalized classical regime where the correlation length is substantial. Figure 1 clearly shows a very short autocorrelation time for the $xy$-plane whereas we detect a considerably longer autocorrelation time for the $z$-direction. This is a direct consequence of the *long* magnetic length scale and the chosen decoupling. The physical reason for the long autocorrelation time corresponds to the restoration of the $SU(2)$ spin symmetry. This insight can be used to define an improved, $SU(2)$ symmetric estimator for the spin-spin

correlation function, namely $(S_{\hat{S}^x} + S_{\hat{S}^y} + S_{\hat{S}^z})/3$. Thereby, global spin rotations are no longer an issue and this improved estimator shows the shortest autocorrelation time as seen clearly in Fig. 1 (b). Other ways to tackle large autocorrelation can be global updates or parallel tempering.

Using the time series of Monte Carlo samples we would like to obtain estimates of the mean and the standard error of the mean. A simple method which we will describe in this tutorial is the rebinning method, also known in the literature as rebatching, where a fixed number (denoted by `N_rebin`) of adjacent original bins are aggregated to form a new effective bin. In addition to measuring the decay rate of the autocorrelation function (44), a measure for the autocorrelation time can be also obtained by the rebinning method. For a comparison to other methods of estimating the autocorrelation time we refer the reader to the literature [82, 83, 87]. A reliable error analysis requires independent bins, otherwise the error is typically underestimated. This behavior is observed in Fig. 1 (b), where the effective bin size has been systematically increased by rebinning. If the effective bin size is smaller than the autocorrelation time the error will be underestimated. When the effective bin size becomes larger than the autocorrelation time converging behavior sets in and in this region the error estimate will be correct.

For the analysis of the Monte Carlo data (see Sec. 3.5), the user can provide a finite value for `N_auto` to trigger the computation of autocorrelation functions $Auto_{\hat{O}}(t_{\text{QMC}})$ in the range $t_{\text{QMC}} = [0, \texttt{N\_auto}]$. Since these computations are quite time consuming and require many Monte Carlo bins the default value is `N_auto=0` if unspecified. To produce Fig. 1, we set $\texttt{N\_auto} = 500$ and used a total of approximately one million bins.

## 2.5 Pseudo code description

---

**Algorithm 1** Basic structure of the auxiliary field QMC implementation in `Prog/main.f90`

---

1: **call** ham_set                                              ▷ *Set the Hamiltonian and the lattice*
2: **call** confin                      ▷ *Read in an auxiliary-field configuration or generate it randomly*

3: **for** $n = L_{\text{Trotter}}$ to 1 **do**        ▷ *Fill the storage, needed for the first actual Monte Carlo sweep*
4:    **call** wrapul        ▷ *Compute propagation matrices and store them at stabilization points*
5: **end for**

6: **for** $n_{\text{bc}} = 1$ to $N_{\text{bin}}$ **do**        ▷ *Loop over bins. The bin defines the unit of Monte Carlo time*

7:    **for** $n_{\text{sw}} = 1$ to $N_{\text{sweep}}$ **do**            ▷ *Loop over sweeps. Each sweep updates twice*
                              ▷ *(upward and downward in imag. time) the space-time lattice of auxiliary fields*

8:       **for** $n_\tau = 1$ to $L_{\text{Trotter}}$ **do**                                    ▷ *Upward sweep*
9:          **call** wrapgrup      ▷ *Propagate Green fct. from $n_{tau} - 1$ to $n_\tau$, and compute new*
                                          ▷ *estimate of Green fct. at $n_\tau$, using sequential updates*

                        ▷ *Stabilization:*
10:          **if** $n_\tau =$ stabilization point in imaginary time **then**
11:             **call** wrapur    ▷ *Compute propagation from previous stabilization point to $n_\tau$*
                 ▷ *Storage management:*
                 ▷ *Read from storage: propagation from $L_{Trotter}$ to $n_\tau$*
                 ▷ *Write to storage : the just computed propagation*
12:             **call** cgr          ▷ *Recalculate the Green function at time $n_\tau$ in a stable way*
13:             **call** control_precisionG     ▷ *Compare propagated and recalculated Green fct.*
14:          **end if**

15:          **if** $n_\tau \in [LOBS\_ST, LOBS\_EN]$ **then**        ▷ *Measure the equal time observables*
16:             **call** obser
17:          **end if**
18:       **end for**

19:       **for** $n_\tau = L_{\text{Trotter}}$ to 1 **do**                                    ▷ *Downward sweep*
                 ▷ *Repeat the above steps (update, propagation, stabilization, equal time*
                 ▷ *measurements) for the downward direction in imaginary time*
20:       **end for**

21:       **if** $n_\tau = 1$ **then**                        ▷ *Measure the time displaced observables*
22:          **call** tau_m
23:       **end if**
24:    **end for**
25:    **call** pr_obs      ▷ *Calculate measurement averages for current bin and write them to disk*
26:    **call** confout                                    ▷ *Write auxiliary-field configuration to disk*
27: **end for**

---

# 3 Data Structures and Input/Output

## 3.1 Implementation of the Hamiltonian and the lattice

The module `Hamiltonian`, contained in the file `Hamiltonian.f90`, defines the model Hamiltonian, the lattice under consideration and the desired observables (Table 2). We have collected a number of example Hamiltonians, lattices and observables in the file `Hamiltonian_Examples.f90`. The examples are described in Sec. 4. To implement a user-defined model, only the module `Hamiltonian` has to be set up. Accordingly, this documentation focusses almost entirely on this module and the subprograms it includes. The remaining parts of the code may hence be treated as a black box.

To specify the Hamiltonian, one needs an `Operator` and a `Lattice` type as well as a type for the observables. These three data structures will be described in the following sections.

| Subprogram | Description | Section |
|---|---|---|
| `Ham_Set` | Reads in model and lattice parameters from the file `parameters` and it sets the Hamiltonian by calling `Ham_latt`, `Ham_hop`, and `Ham_V`. | |
| `Ham_hop` | Sets the hopping term $\hat{\mathscr{H}}_T$ by calling `Op_make` and `Op_set`. | 3.1.1, 3.1.2 |
| `Ham_V` | Sets the interaction terms $\hat{\mathscr{H}}_V$ and $\hat{\mathscr{H}}_I$ by calling `Op_make` and `Op_set`. | 3.1.1, 3.1.2 |
| `Ham_Latt` | Sets the lattice by calling `Make_Lattice`. | 3.1.3 |
| `S0` | A function which returns an update ratio for the Ising term $\hat{\mathscr{H}}_{I,0}$. | 4.4.3 |
| `Alloc_obs` | Assigns memory storage to the observables | |
| `Obser` | Computes the scalar observables and equal-time correlation functions. | 3.2 |
| `ObserT` | Computes time-displaced correlation functions. | 3.2 |
| `Init_obs` | Initializes the observables to zero. | |
| `Pr_obs` | Writes the observables to the disk by calling `Print_bin`. | |

Table 2: Overview of the subprograms of the module `Hamiltonian` to define the Hamiltonian, the lattice and the observables. The highlighted subroutines have to be modified by the user.

### 3.1.1 The `Operator` type

The fundamental data structure in the code is the data structure `Operator`. It is implemented as a Fortran derived data type. This type is used to define the Hamiltonian (2). In general, the matrices $\mathbf{T}^{(ks)}$, $\mathbf{V}^{(ks)}$ and $\mathbf{I}^{(ks)}$ are sparse Hermitian matrices. Consider the matrix $X$ of dimension $N_{\text{dim}} \times N_{\text{dim}}$, as a representative for each of the above three matrices. Let us denote with $\{z_1, \cdots, z_N\}$ a subset of $N$ indices, for which

$$X_{x,y} \begin{cases} \neq 0 & \text{if } x, y \in \{z_1, \cdots z_N\} \\ = 0 & \text{otherwise} \end{cases} \tag{45}$$

Usually, we have $N \ll N_{\text{dim}}$. We define the $N \times N_{\text{dim}}$ matrices $\mathbf{P}$ as

$$P_{i,x} = \delta_{z_i,x} \,, \tag{46}$$

where $i \in [1, \cdots, N]$ and $x \in [1, \cdots, N_{\text{dim}}]$. The matrix $\mathbf{P}$ selects the non-vanishing entries of $X$, which are contained in the rank-$N$ matrix $O$:

$$X = P^T O P \,, \tag{47}$$

and

$$X_{x,y} = \sum_{i,j}^{N} P_{i,x} O_{i,j} P_{j,y} = \sum_{i,j}^{N} \delta_{z_i,x} O_{ij} \delta_{z_j,y} \,. \tag{48}$$

Since the $\boldsymbol{P}$ matrices have only one non-vanishing entry per column, they can conveniently be stored as a vector $\vec{P}$, with entries

$$P_i = z_i. \tag{49}$$

There are many useful identities which emerge from this structure. For example:

$$e^X = e^{\boldsymbol{P}^T \boldsymbol{O} \boldsymbol{P}} = \sum_{n=0}^{\infty} \frac{\left(\boldsymbol{P}^T \boldsymbol{O} \boldsymbol{P}\right)^n}{n!} = \mathbb{1} + \boldsymbol{P}^T \left(e^{\boldsymbol{O}} - \mathbb{1}\right) \boldsymbol{P} \,, \tag{50}$$

since

$$\boldsymbol{P} \boldsymbol{P}^T = \mathbb{1}_{N \times N}. \tag{51}$$

In the code, we define a structure called `Operator` to capture the above. This type `Operator` bundles several components that are needed to define and use an operator matrix in the program.

### 3.1.2 Specification of the model

| Variable | Type | Description |
|---|---|---|
| Op_X%N | Integer | Effective dimension $N$ |
| Op_X%O | Complex | Matrix $\mathbf{O}$ of dimension $N \times N$ |
| Op_X%P | Integer | Matrix $\mathbf{P}$ encoded as a vector of dimension $N$ |
| Op_X%g | Complex | Coupling strength $g$ |
| Op_X%alpha | Complex | Constant $\alpha$ |
| Op_X%type | Integer | Parameter to set the type of HS transformation (1 = Ising, 2 = discrete HS for perfect-square term) |
| Op_X%U | Complex | Matrix containing the eigenvectors of $\mathbf{O}$ |
| Op_X%E | Real | Eigenvalues of $\mathbf{O}$ |
| Op_X%N_non_zero | Integer | Number of non-vanishing eigenvalues of $\mathbf{O}$ |

Table 3: Member variables of the `Operator` type. In the left column, the letter X is a placeholder for the letters T and V, indicating hopping and interaction operators, respectively. The highlighted variables have to be specified by the user.

In this section we show how to specify the Hamiltonian (2) in the code. More precisely, we have to set the matrix representation of the imaginary-time propagators – $e^{-\Delta\tau T^{(ks)}}$, $e^{\sqrt{-\Delta\tau U_k} \eta_{k\tau} V^{(ks)}}$, and $e^{-\Delta\tau s_{k\tau} I^{(ks)}}$ – that appear in the partition function (15). For each pair of indices $(k,s)$, these terms have the general form

$$\text{Matrix Exponential} = e^{g\,\phi(\texttt{type})\mathbf{X}} \,. \tag{52}$$

In case of the perfect-square term, we additionally have to set the constant $\alpha$, see the definition of the operators $\hat{V}^{(k)}$ in Eq. (4). The data structures which hold all the above information are variables of the type `Operator` (see Table 3). For each pair of indices $(k,s)$, we store the following parameters in an `Operator` variable:

- $\vec{P}$ and $\mathbf{O}$ defining the matrix $\mathbf{X}$ [see Eq. (47)]

- the constants $g$, $\alpha$

- optionally: the type `type` of the discrete fields $\phi$

In case of the Ising term, we store `type=1` which sets $\phi_{k\tau} = s_{k\tau}$. In case of the perfect-square term, the field results from the discrete HS transformation (10) and we store `type=2` which sets $\phi_{k\tau} = \eta_{k\tau}$. Note that we have dropped the color index $\sigma$, since the implementation uses the $SU(N_{\text{col}})$ invariance of the Hamiltonian.

Accordingly, the following data structures fully describe the Hamiltonian (2):

- For the hopping Hamiltonian (3), we have to set the exponentiated hopping matrices $e^{-\Delta\tau T^{(ks)}}$:

  In this case $\mathbf{X}^{(ks)} = \mathbf{T}^{(ks)}$. Precisely, a single variable `Op_T` describes the operator matrix

  $$\left( \sum_{x,y}^{N_{\text{dim}}} \hat{c}_x^\dagger T_{xy}^{(ks)} \hat{c}_y \right), \tag{53}$$

  where $k = [1, M_T]$ and $s = [1, N_{\text{fl}}]$. To make contact with the general expression (52) we set $g = -\Delta\tau$ (and $\alpha = 0$). In case of the hopping matrix, the type variable `Op_T%type` is neglected by the code. All in all, the corresponding array of structure variables is `Op_T(M_T,N_fl)`.

- For the interaction Hamiltonian (4), which is of perfect-square type, we have to set the exponentiated matrices $e^{\sqrt{-\Delta\tau U_k}\eta_{k\tau} V^{(ks)}}$:

  In this case, $\mathbf{X} = \mathbf{V}^{(ks)}$. A single variable `Op_V` describes the operator matrix:

  $$\left[ \left( \sum_{x,y}^{N_{\text{dim}}} \hat{c}_x^\dagger V_{x,y}^{(ks)} \hat{c}_y \right) + \alpha_{ks} \right], \tag{54}$$

  where $k = [1, M_V]$ and $s = [1, N_{\text{fl}}]$. To make contact with the general expression (52) and to set the constant $\alpha$, we choose $g = \sqrt{-\Delta\tau U_k}$ and $\alpha = \alpha_{ks}$. The discrete Hubbard-Stratonovich decomposition which is used for the perfect-square interaction, is selected by setting the type variable to `Op_V%type = 2`. All in all, the required structure variables `Op_V` are defined using the array `Op_V(M_V,N_fl)`.

- For the Ising interaction Hamiltonian (5), we have to set the exponentiated matrices $e^{-\Delta\tau s_{k\tau} I^{(ks)}}$:

  In this case, $\mathbf{X} = \mathbf{I}^{(k,s)}$. A single variable `Op_V` then describes the operator matrix:

  $$\left( \sum_{x,y}^{N_{\text{dim}}} \hat{c}_x^\dagger I_{xy}^{(ks)} \hat{c}_y \right), \tag{55}$$

  where $k = [1, M_I]$ and $s = [1, N_{\text{fl}}]$. To make contact with the general expression (52), we set $g = -\Delta\tau$ (and $\alpha = 0$). The Ising interaction is specified by setting the type variable `Op_V%type=1`. All in all, the required structure variables are contained in the array `Op_V(M_I,N_fl)`.

- In case of a full interaction [perfect-square term (4) and Ising term (5)], we define the corresponding doubled array `Op_V(M_V+M_I,N_fl)` and set the variables separately for both ranges of the array according to the above.

### 3.1.3 The `Lattice` type

We have a lattice module which can generate one- and two-dimensional Bravais lattices. Note that the orbital structure of each unit cell has to be specified by the user in the Hamiltonian module. The user has to specify unit vectors $\vec{a}_1$ and $\vec{a}_2$ as well as the size of the lattice. The size is characterized by two vectors $\vec{L}_1$ and $\vec{L}_2$ and the lattice is placed on a torus (periodic boundary conditions):

$$\hat{c}_{\vec{i}+\vec{L}_1} = \hat{c}_{\vec{i}+\vec{L}_2} = \hat{c}_{\vec{i}}. \tag{56}$$

The function call

```
Call Make_Lattice( L1, L2, a1,  a2, Latt )
```

will generate the lattice `Latt` of type `Lattice`. Note again that the orbital structure of the unit cell has to be provided by the user. The reciprocal lattice vectors are defined by:

$$\vec{a}_i \cdot \vec{g}_i = 2\pi \delta_{i,j}, \tag{57}$$

and the Brillouin zone corresponds to the Wigner-Seitz cell of the lattice. With $\vec{k} = \sum_i \alpha_i \vec{g}_i$, the k-space quantization follows from:

$$\begin{bmatrix} \vec{L}_1 \cdot \vec{g}_1 & \vec{L}_1 \cdot \vec{g}_2 \\ \vec{L}_2 \cdot \vec{g}_1 & \vec{L}_2 \cdot \vec{g}_2 \end{bmatrix} \begin{bmatrix} \alpha_1 \\ \alpha_2 \end{bmatrix} = 2\pi \begin{bmatrix} n \\ m \end{bmatrix} \tag{58}$$

such that

$$\vec{k} = n\vec{b}_1 + m\vec{b}_2, \quad \text{with}$$

$$\vec{b}_1 = \frac{2\pi}{(\vec{L}_1 \cdot \vec{g}_1)(\vec{L}_2 \cdot \vec{g}_2) - (\vec{L}_1 \cdot \vec{g}_2)(\vec{L}_2 \cdot \vec{g}_1)} \left[ (\vec{L}_2 \cdot \vec{g}_2)\vec{g}_1 - (\vec{L}_2 \cdot \vec{g}_1)\vec{g}_2 \right], \quad \text{and}$$

$$\vec{b}_2 = \frac{2\pi}{(\vec{L}_1 \cdot \vec{g}_1)(\vec{L}_2 \cdot \vec{g}_2) - (\vec{L}_1 \cdot \vec{g}_2)(\vec{L}_2 \cdot \vec{g}_1)} \left[ (\vec{L}_1 \cdot \vec{g}_1)\vec{g}_2 - (\vec{L}_1 \cdot \vec{g}_2)\vec{g}_1 \right]. \tag{59}$$

The `Lattice` module equally handles the Fourier transformation. For example the subroutine `Fourier_R_to_K` carries out the transformation:

$$S(\vec{k}, :, :, :) = \frac{1}{N_{\text{unit cell}}} \sum_{\vec{i}, \vec{j}} e^{-i\vec{k} \cdot (\vec{i} - \vec{j})} S(\vec{i} - \vec{j}, :, :, :), \tag{60}$$

and `Fourier_K_to_R` the inverse Fourier transform

$$S(\vec{r}, :, :, :) = \frac{1}{N_{\text{unit cell}}} \sum_{\vec{k} \in BZ} e^{i\vec{k} \cdot \vec{r}} S(\vec{k}, :, :, :). \tag{61}$$

In the above, the unspecified dimensions of the structure factor can refer to imaginary-time and orbital indices.

## 3.2 The observable types `Obser_Vec` and `Obser_Latt`

Our definition of the model includes observables [Eq. (24)]. We have defined two observable types: `Obser_vec` for an array of scalar observables such as the energy, and `Obser_Latt` for correlation functions that have the lattice symmetry. In the latter case, translation symmetry can be used to provide improved estimators and to reduce the size of the output. We also obtain improved estimators by taking measurements in the imaginary-time interval [LOBS_ST,LOBS_EN] (see the parameter file in Sec. 3.3.1) thereby exploiting the invariance

| Variable | Type | Description |
|---|---|---|
| `Latt%a1_p, Latt%a2_p` | Real | Unit vectors $\vec{a}_1$, $\vec{a}_2$ |
| `Latt%L1_p, Latt%L2_p` | Real | Vectors $\vec{L}_1$, $\vec{L}_2$ that define the topology of the lattice. Tilted lattices are thereby possible to implement. |
| `Latt%N` | Integer | Number of lattice points, $N_{\text{unit cell}}$ |
| `Latt%list` | Integer | Maps each lattice point $i = 1, \cdots, N_{\text{unit cell}}$ to a real space vector denoting the position of the unit cell: $\vec{R}_i = \texttt{list(i,1)} \, \vec{a}_1 + \texttt{list(i,2)} \, \vec{a}_2 \equiv i_1 \vec{a}_1 + i_2 \vec{a}_2$ |
| `Latt%invlist` | Integer | $\texttt{Invlist}(i_1, i_2) = i$ |
| `Latt%nnlist` | Integer | $j = \texttt{nnlist}(i, n_1, n_2)$, $n_1, n_2 \in [-1, 1]$ $\vec{R}_j = \vec{R}_i + n_1 \vec{a}_1 + n_2 \vec{a}_2$ |
| `Latt%imj` | Integer | $\vec{R}_{imj(i,j)} = \vec{R}_i - \vec{R}_j$. $imj, i, j \in 1, \cdots, N_{\text{unit cell}}$ |
| `Latt%BZ1_p, Latt%BZ2_p` | Real | Reciprocal space vectors $\vec{g}_i$ (See Eq. 57) |
| `Latt%b1_p, Latt%b1_p` | Real | $k$-quantization (See Eq. 59) |
| `Latt%listk` | Integer | Maps each reciprocal lattice point $k = 1, \cdots, N_{\text{unit cell}}$ to a reciprocal space vector $\vec{k}_k = \texttt{listk(k,1)} \, \vec{b}_1 + \texttt{listk(k,2)} \, \vec{b}_2 \equiv k_1 \vec{b}_1 + k_2 \vec{b}_2$ |
| `Latt%invlistk` | Integer | $\texttt{Invlistk}(k_1, k_2) = k$ |
| `Latt%b1_perp_p, Latt%b2_perp_p` | Real | Orthonormal vectors to $\vec{b}_i$. For internal use. |

Table 4: Components of the `Lattice` type for two-dimensional lattices using as example the default lattice name `Latt`. The highlighted variables have to be specified by the user. Other components of the `Lattice` are generated upon calling: `Call Make_Lattice( L1, L2, a1, a2, Latt )`.

under translation in imaginary-time. Note that the translation symmetries in space and in time are *broken* for a given configuration $C$ but restored by the Monte Carlo sampling. In general, the user defines the size and the number of bins in the parameter file, each bin having a given amount of sweeps. Within a sweep we run sequentially through the HS and Ising fields, from time slice 1 to time slice $L_{\text{Trotter}}$ and back. The results of each bin are written to a file and analyzed at the end of the run.

To accomplish the reweighting of observables (see Sec. 2.1.3), for each configuration the measured value of an observable is multiplied by the factors ZS and ZP:

$$\text{ZS} = \text{sign}(C) \,, \tag{62}$$

$$\text{ZP} = \frac{e^{-S(C)}}{\Re\left[e^{-S(C)}\right]} \,. \tag{63}$$

They are computed from the Monte Carlo phase of a configuration,

$$\texttt{phase} = \frac{e^{-S(C)}}{\left|e^{-S(C)}\right|} \,, \tag{64}$$

which is provided by the main program. Note that each observable structure also includes the average sign [Eq. (25)].

### 3.2.1 Scalar observables

This data type is described in Table 5 and is useful to compute an array of scalar observables. Consider a variable `Obs` of type `Obser_vec`. At the beginning of each bin, a call to

`Obser_Vec_Init` in the module `observables_mod.f90` will set `Obs%N=0`, `Obs%Ave_sign` `=0` and `Obs%Obs_vec(:)=0`. Each time the main program calls the routine `Obser` in the Hamiltonian module, the counter `Obs%N` is incremented by one, the sign (see Eq. 23) is accumulated in the variable `Obs%Ave_sign`, and the desired observables (multiplied by the sign and $\frac{e^{-S(C)}}{\Re\left[e^{-S(C)}\right]}$, see Sec. 2.1.2) are accumulated in the vector `Obs%Obs_vec`. At the end of

| Variable | Type | Description | Contribution of configuration $C$ |
|---|---|---|---|
| `Obs%N` | Int. | Number of measurements | |
| `Obs%Ave_sign` | Real | Cumulated sign [Eq. (25)] | sign($C$) |
| `Obs%Obs_vec(:)` | Compl. | Cumulated vector of observables [Eq. (24)] | $\langle\langle \hat{O}(:)\rangle\rangle_C \frac{e^{-S(C)}}{\Re\left[e^{-S(C)}\right]}$ sign $(C)$ |
| `Obs%File_Vec` | Char. | Name of output file | |

Table 5: Components of the `Obser_vec` type. The table lists the data included in a variable Obs of type `Obser_vec`.

the bin, a call to `Print_bin_Vec` in module `observables_mod.f90` will append the result of the bin in the file `File_Vec_scal`. Note that this subroutine will automatically append the suffix *_scal* to the the filename `File_Vec`. This suffix is important to allow automatic analysis of the data at the end of the run.

### 3.2.2 Equal time and time displaced correlation functions

| Variable | Type | Description | Contribution of configuration $C$ |
|---|---|---|---|
| `Obs%N` | Int. | Number of measurements | |
| `Obs%Ave_sign` | Real | Cumulated sign [Eq. (25)] | sign($C$) |
| `Obs%Obs_latt` $(\vec{i}-\vec{j}, \tau, \alpha, \beta)$ | Compl. | Cumululated correlation function [Eq. (24)] | $\langle\langle \hat{O}_{\vec{i},\alpha}(\tau)\hat{O}_{\vec{j},\beta}\rangle\rangle_C \frac{e^{-S(C)}}{\Re\left[e^{-S(C)}\right]}$sign($C$) |
| `Obs%Obs_latt0`$(\alpha)$ | Compl. | Cumulated expectation value [Eq. (24)] | $\langle\langle \hat{O}_{\vec{i},\alpha}\rangle\rangle_C \frac{e^{-S(C)}}{\Re\left[e^{-S(C)}\right]}$ sign $(C)$ |
| `Obs%File_Latt` | Char. | Name of output file | |

Table 6: Components of the `Obser_latt` type. The table lists the data included in a variable Obs of type `Obser_latt`.

This data type (see Table 6) is useful so as to deal with equal time as well as imaginary-time displaced correlation functions of the form:

$$S_{\hat{O},\alpha,\beta}(\vec{k},\tau) = \frac{1}{N_{\text{unit cell}}} \sum_{\vec{i},\vec{j}} e^{-\vec{k}\cdot(\vec{i}-\vec{j})} \left( \langle \hat{O}_{\vec{i},\alpha}(\tau)\hat{O}_{\vec{j},\beta}\rangle - \langle \hat{O}_{\vec{i},\alpha}\rangle\langle \hat{O}_{\vec{j},\beta}\rangle \right). \tag{65}$$

Here, translation symmetry of the Bravais lattice is explicitly taken into account. The correla-

tion function splits in a correlated part $S^{(\text{corr})}_{\hat{O},\alpha,\beta}(\vec{k},\tau)$ and a background part $S^{(\text{back})}_{\hat{O},\alpha,\beta}(\vec{k})$:

$$S^{(\text{corr})}_{\hat{O},\alpha,\beta}(\vec{k},\tau) = \frac{1}{N_{\text{unit cell}}} \sum_{\vec{i},\vec{j}} e^{-i\vec{k}\cdot(\vec{i}-\vec{j})} \langle \hat{O}_{\vec{i},\alpha}(\tau)\hat{O}_{\vec{j},\beta} \rangle \,, \tag{66}$$

$$S^{(\text{back})}_{\hat{O},\alpha,\beta}(\vec{k}) = \frac{1}{N_{\text{unit cell}}} \sum_{\vec{i},\vec{j}} e^{-i\vec{k}\cdot(\vec{i}-\vec{j})} \langle \hat{O}_{\vec{i},\alpha}(\tau) \rangle \langle \hat{O}_{\vec{j},\beta} \rangle$$

$$= N_{\text{unit cell}} \langle \hat{O}_{\alpha} \rangle \langle \hat{O}_{\beta} \rangle \, \delta(\vec{k}) \,, \tag{67}$$

where translation invariance in space and time has been exploited to obtain the last line. The background part depends only on the expectation value $\langle \hat{O}_{\alpha} \rangle$, for which we use the following estimator

$$\langle \hat{O}_{\alpha} \rangle \equiv \frac{1}{N_{\text{unit cell}}} \sum_{\vec{i}} \langle \hat{O}_{\vec{i},\alpha} \rangle \,. \tag{68}$$

Consider a variable Obs of type Obser_latt. At the beginning of each bin a call to the subroutine Obser_Latt_Init in the module observables_mod.f90 will initialize the elements of Obs to zero. Each time the main program calls the Obser or ObserT routines one accumulates the quantity $\langle\langle \hat{O}_{\vec{i},\alpha}(\tau)\hat{O}_{\vec{j},\beta} \rangle\rangle_C \frac{e^{-S(C)}}{\Re[e^{-S(C)}]} \text{sign}(C)$ in Obs%Obs_latt$(\vec{i}-\vec{j},\tau,\alpha,\beta)$ and $\langle\langle \hat{O}_{\vec{i},\alpha} \rangle\rangle_C \frac{e^{-S(C)}}{\Re[e^{-S(C)}]} \text{sign}(C)$ in Obs%Obs_latt0$(\alpha)$. At the end of each bin, a call to Print_bin_Latt in the module observables_mod.f90 will append the result of the bin in the specified file Obs%File_Latt. Note that the routine Print_bin_Latt carries out the Fourier transformation and prints the results in $k$-space. We have adopted the following naming conventions. For equal time observables, defined by having the second dimension of the array Obs%Obs_latt$(\vec{i}-\vec{j},\tau,\alpha,\beta)$ set to unity, the routine Print_bin_Latt attaches the suffix _eq to Obs%File_Latt. For time displaced correlation functions we use the suffix _tau.

## 3.3 File structure

| Directory | Description |
|---|---|
| Prog/ | Main program and subroutines |
| Libraries/ | Collection of mathematical routines |
| Analysis/ | Routines for error analysis |
| Examples/ | Example simulations for Hubbard-type models |
| Start/ | Parameter files and scripts |
| Documentation/ | Documentation of the QMC code. |

Table 7: Overview of the directories.

The code package consists of the program directories Prog/, Libraries/ and Analysis/. The example simulations corresponding to the walkthroughs of Sec. 4.1 - 4.4 are included in Examples/. The package content is summarized in Table 7.

### 3.3.1 Input files

The input files are listed in Table 8. To enable restarting a previous simulation (see Table 10) or to use a given HS and Ising field configuration as input for a new simulation, the program reads in the files confin_<threadnumber> in case they are present. It goes without saying that the dimensions of the thereby defined field configuration (number of threads, lattices size, and number of time slices) have to match the corresponding values of the parameter file. The

| File | Description |
|------|-------------|
| `parameters` | This collects input data. We can set here the parameters for the lattice, which model, variables of the QMC process, and the error analysis. |
| `seeds` | List of integer numbers to initialize the random number generator and to start a simulation from scratch. |
| `(confin_<threadnumber>)` | (Optionally, a HS and Ising field configuration can be provided as input.) |

Table 8: Overview of the input files in `Start/` required for a simulation.

parameter file `Start/parameters` has the following form – using as an example the $SU(2)$-symmetric Hubbard model on a square lattice (see Sec. 4.1 for a detailed walkthrough):

```
!==========================================================================
!  Variables for the Hubb program
!--------------------------------------------------------------------------
&VAR_lattice
L1 = 4                    ! Length in direction a_1
L2 = 4                    ! Length in direction a_2
Lattice_type = "Square"   ! a_1 = (1,0),a_2=(0,1),  Norb=1, N_coord=2
!Lattice_type ="Honeycomb"! a_1 = (1,0),a_2 =(1/2,sqrt(3)/2),Norb=2,N_coord=3
Model = "Hubbard_SU2"     ! Sets Nf=1, N_sun=2. HS field couples to the
                          ! density
!Model = "Hubbard_Mz"     ! Sets Nf=2, N_sun=1. HS field couples to the
                          ! z-component of magnetization.
!Model="Hubbard_SU2_Ising"! Sets Nf_1, N_sun=2 and runs only for the square
                          ! lattice
                          ! Hubbard model  coupled to transverse Ising field
/
&VAR_Hubbard              ! Variables for the Hubbard model
ham_T   = 1.D0            ! Hopping parameter
ham_chem= 0.D0            ! chemical potential
ham_U   = 4.D0            ! Hubbard interaction
Beta    = 5.D0            ! inverse temperature
dtau    = 0.1D0           ! Thereby Ltrot=Beta/dtau
/

&VAR_Ising                ! Model parameters for the Ising code
Ham_xi = 1.d0             ! Only needed if Model="Hubbard_SU2_Ising"
Ham_J  = 0.2d0
Ham_h  = 2.d0
/

&VAR_QMC                  ! Variables for the QMC run
Nwrap   = 10              ! Stabilization. Green functions is computed from
                          ! scratch after each time interval  Nwrap*Dtau
NSweep  = 500             ! Number of sweeps
NBin    = 2               ! Number of bins
Ltau    = 1               ! 1 for calculation of time displ. Green functions;
                          ! 0 otherwise
LOBS_ST = 1               ! Start measurements at time slice LOBS_ST
LOBS_EN = 50              ! End   measurements at time slice LOBS_EN
CPU_MAX = 0.1             ! Code will stop after CPU_MAX hours.
                          ! If not specified, code will stop after Nbin bins.
/
&VAR_errors               ! Variables for analysis programs
```

```
n_skip  = 1              ! Number of bins that will be skipped.
N_rebin = 1              ! Rebinning
N_Cov   = 0              ! If set to 1 covariance will be computed
                        ! for unequal time correlation functions.
N_auto  = 100            ! If set to >0 autocorrelation function will be
                        ! computed for scalar and equal time observables.
/
```

### 3.3.2 Output: Observables

| File | Description |
|------|-------------|
| info | After completion of the simulation, this file documents parameters of the model, the QMC run and simulation metrics (precision, acceptance rate, wallclock time). |
| X_scal | Results of equal time measurements of scalar observables. The placeholder X stands for the observables Kin, Pot, Part, and Ener. |
| Y_eq, Y_tau | Results of equal time and time displaced measurements of correlation functions. The placeholder Y stands for Green, SpinZ, SpinXY, and Den. |
| confout_<threadnumber> | Output files for the HS and Ising field configuration. |

Table 9: Overview of the standard output files. See Sec. 3.2 for the definitions of observables and correlation functions.

The standard output files are listed in Table 9. The output of the measured data is organized in bins. One bin corresponds to the arithmetic average over a fixed number of individual measurements which depends on the chosen measurement interval [LOBS_ST, LOBS_EN] on the imaginary-time axis and on the number NSweep of Monte Carlo sweeps. If the user runs an MPI parallelized version of the code, the average also extends over the number of MPI threads. The formatting of the output for a single bin depends on the observable type, Obs_vec or Obs_Latt:

- Observables of type Obs_vec: For each additional bin, a single new line is added to the output file. In case of an observable with N_size components, the formatting is

  ```
  N_size+1  <measured value,1> ... <measured value,N_size>
     <measured sign>
  ```

  The counter variable N_size+1 refers to the number of measurements per line, including the phase measurement. This format is required by the error analysis routine (see Sec. 3.5). Scalar observables like kinetic energy, potential energy, total energy and particle number are treated as a vector of size N_size=1.

- Observables of type Obs_Latt: For each additional bin, a new data block is added to the output file. The block consists of the expectation values [Eq. (68)] contributing to the background part [Eq. (67)] of the correlation function, and the correlated part [Eq. (66)] of the correlation function. For imaginary-time displaced correlation functions, the formatting of the block follows this scheme:

  ```
  <measured sign> <N_orbital> <N_unit_cell> <N_time_slices> <dtau>
  do alpha = 1, N_orbital
  ```

```
        ⟨Ô_α⟩
    enddo
    do i = 1, N_unit_cell
        <reciprocal lattice vector k(i)>
        do tau = 1, N_time_slices
            do alpha = 1, N_orbital
                do beta = 1, N_orbital
                    ⟨S^(corr)_{Ô,α,β}(k(i),τ)⟩
                enddo
            enddo
        enddo
    enddo
```

The same block structure is used for equal time correlation functions, except for the entries `<N_time_slices>` and `<dtau>` which are not present in the latter. Using this structure for the bins as input, the full correlation function $S_{\hat{O},\alpha,\beta}(\vec{k},\tau)$ [Eq. (65)] is then calculated by calling the error analysis routine (see Sec. 3.5).

### 3.3.3 Output: Precision

The finite temperature auxiliary field QMC algorithm is known to be numerically unstable, as discussed in Sec. 2.3. The origin the numerical instabilities arises from the imaginary-time propagation which invariably leads to exponentially small and exponentially large scales. Numerical stabilization of the code is delicate and has been pioneered in Ref. [2] for the finite-temperature algorithm and in Refs. [3,4] for the zero temperature projective algorithm. As shown in Ref. [7] scales can be omitted in the ground state algorithm – thus rendering it very stable – but have to be taken into account in the finite-temperature code. Apart from runtime information, the file `info` contains important information concerning the stability of the code. It is important to know that numerical stabilization is delicate and there is no guarantee that it will work for all models.

If the numerical stabilization turns out to be bad, one option is to reduce the value of the parameter `Nwrap` in the parameter file. For performing the stabilization of the involved matrix multiplications we rely on routines from LAPACK. Hence it is very likely that your results may change significantly if you switch the LAPACK implementation. In order to offer a simple baseline to which people can quickly switch if they want to see whether their results depend on the library used for linear algebra routines we have included parts of the LAPACK-3.6.1 reference implementation from http://www.netlib.org/lapack/. You can switch to the QR decomposition related routines from the LAPACK reference implementation by including the switch -DQRREF into the PROGRAMCONFIGURATION string. To use these routines you need to link against a lapack library that implements at least the LAPACK-3.4.0 interface.[5]

To provide further flexibility, we have kept the history of different stabilization schemes. Our default strategy is quick and generically works well but we have encountered some models where it fails. If this applies to your model, you can use the switch -DSTAB2 (stabilization scheme based on the QR decomposition, but not using the LAPACK reference implementation) or -DSTAB1 (stabilization scheme based on singular value decomposition) in the header of the file `Makefile` and recompile the code.

Typical values for the numerical precision can be found in the examples of Sec. 4 (see Sec. 4.1.3 and 4.2.3).

---

[5] We have encountered some compiling issues with this flag. In particular the older intel ifort compiler version 10.1 fails for all optimization levels.

## 3.4 Scripts

| Script | Description | Section |
|---|---|---|
| `setenv.sh` | Exports the path variable. | 3.6 |
| `Start/analysis.sh` | Starts the error analysis. | 3.5, 3.6 |
| `Start/out_to_in.sh` | Copies the output configurations of HS and Ising spins to the respective input files. | 3.6 |

Table 10: Overview of the bash script files.

## 3.5 Analysis programs

| Program | Description |
|---|---|
| `cov_scal.f90` | In combination with the script `analysis.sh`, the bin files with suffix `_scal` are read in, and the corresponding files with suffix `_scalJ` are produced. They contain the result of the Jackknife rebinning analysis (see Sec. 2.4). |
| `cov_eq.f90` | In combination with the script `analysis.sh`, the bin files with suffix `_eq` are read in, and the corresponding files with suffix `_eqJR` and `_eqJK` are produced. They correspond to correlation functions in real and Fourier space, respectively. |
| `cov_tau.f90` | In combination with the script `analysis.sh`, the bin files `X_tau` are read in, and the directories `X_kx_ky` are produced for all `kx` and `ky` greater or equal to zero. Here `X` is a place holder from `Green`, `SpinXY`, etc as specified in `Alloc_obs(Ltau)` (See section 4.1.2). Each directory contains a file `g_kx_ky` containing the time displaced correlation function traced over the orbitals. It also contains the covariance matrix if `N_cov` is set to unity in the parameter file (see Sec. 3.3.1). Equally, a directory `X_R0` for the local time displaced correlation function is generated. |

Table 11: Overview of analysis programs that are called within the script `analysis.sh`.

Here we briefly discuss the analysis programs which read in bin/s and carry out the error analysis. (See Sec. 2.4 for a more detailed discussion.) Error analysis is based on the central limit theorem, which requires bins to be statistically independent, and also the existence of a well-defined variance for the observable under consideration. The former will be the case if bins are longer than the autocorrelation time. The latter has to be checked by the user. In the parameter file listed in Sec. 3.3.1, the user can specify how many initial bins should be omitted (variable `n_skip`). This number should be at least comparable or lager than the autocorrelation time. The analysis of the autocorrelation time is triggered by specifying a positive value for `N_auto` that is turned off be default (`N_auto = 0`). The rebinning variable `N_rebin` will merge `N_rebin` bins into a single new bin. If the autocorrelation time is smaller than the effective bin size, the error should become independent of the bin size and thereby of the variable `N_rebin`. Our analysis is based on the Jackknife resampling [57,85], which includes proper treatment of the sign. As listed in Table 11 we provide three analysis programs to account for the three observable types. The programs can be found in the directory `Analysis` and are executed by running the bash shell script `analysis.sh`. In the following, we describe the formatting of the output files mentioned in Table 13.

| File | Description |
| --- | --- |
| `parameters` | Contains also variables for the error analysis: `n_skip`, `N_rebin`, `N_Cov` and `N_auto` (see Sec. 3.3.1) |
| `X_scal, Y_eq, Y_tau` | Monte Carlo bins (see Table 9) |

Table 12: Standard input files for the error analysis.

| File | Description |
| --- | --- |
| `X_scalJ` | Jackknife mean and error of X, where X stands for `Kin`, `Pot`, `Part`, and `Ener`. |
| `X_scal_Auto_N` | QMC-time resolved autocorrelation and rebinning analysis of X, where X stands for `Kin`, `Pot`, `Part`, and `Ener` and N labels the component if X is a vector. |
| `Y_eqJR and Y_eqJK` | Jackknife mean and error of Y, where Y stands for `Green`, `SpinZ`, `SpinXY`, and `Den`. The suffixes R and K refer to real and reciprocal space, respectively. |
| `Y_R0/g_R0` | Time-resolved and spatially local Jackknife mean and error of Y, where Y stands for `Green`, `SpinZ`, `SpinXY`, and `Den`. |
| `Y_eq_Auto_Tr_kx_ky` | QMC-time resolved autocorrelation and rebinning analysis of Y, where Y stands for `Green`, `SpinZ`, `SpinXY`, and `Den`. |
| `Y_kx_ky/g_kx_ky` | Time resolved and $\vec{k}$-dependent Jackknife mean and error of Y, where Y stands for `Green`, `SpinZ`, `SpinXY`, and `Den`. |

Table 13: Standard output files of the error analysis.

- For the scalar quantities X, the output files `X_scalJ` have the following formatting:

```
Effective number of bins, and bins:   <N_bin - n_skip>   <N_bin>

OBS :   1      <mean(X)>      <error(X)>

OBS :   2      <mean(sign)>   <error(sign)>
```

- For the autocorrelation analysis of scalar quantities X, the output files `X_scal_Auto_N` have the following formatting:

```
do i = 1, N_auto
   tau(i)/n_rebin   Auto_X(tau)   <error( X )>
enddo
```

- For the equal time correlation functions Y, the formatting of the output files `Y_eqJR` and `Y_eqJK` follows this structure:

```
do i = 1, N_unit_cell
   <k_x(i)>   <k_y(i)>
   do alpha = 1, N_orbital
   do beta  = 1, N_orbital
     alpha  beta  Re<mean(Y)>  Re<error(Y)>  Im<mean(Y)>  Im<
   error(Y)>
   enddo
```

```
        enddo
     enddo
```

where `Re` and `Im` refer to the real and imaginary part, respectively.

- For the autocorrelation analysis of equal time quantities `Y`, the output files `Y_eq_Auto_Tr_kx_ky` have the following formatting:

```
do i = 1, N_auto
    tau(i)/n_rebin   Auto_Tr[Y](tau)   <error( Tr[Y] )>
enddo
```

- The imaginary-time displaced correlation functions `Y` are written to the output files `Y_R0/g_R0`, when measured locally in space, and to the output files `Y_kx_ky/g_kx_ky` when they are measured $\vec{k}$-resolved. Both output files have the following formatting:

```
do i = 0, Ltau
    tau(i)   <mean( Tr[Y] )>   <error( Tr[Y])>
enddo
```

where `Tr` corresponds to the trace over the orbital degrees of freedom.

## 3.6 Running the code

In this section we describe the steps how to compile and run the code, as well as how to perform the error analysis of the data.

### 3.6.1 Compilation

The environment variables and the directives to compile the code are set in the following makefile `Makefile`:

```
# -DMPI selects MPI.
# -DSTAB1   Alternative stabilization, using the singular value decomposition.
# -DSTAB2   Alternative stabilization, lapack QR with  manual pivoting.
#           Packed form of QR factorization is not used.
# (no flag) Default  stabilization, using lapack QR with pivoting.
#           Packed form of QR factorization  is used.
# -DQRREF   Enables reference lapack implementation of QR decomposition.
# Recommendation: just use the -DMPI flag if you want to run in parallel or
#                 leave it empty for serial jobs.
#                 The default stabilization, no flag, is generically the best.
PROGRAMCONFIGURATION = -DMPI
PROGRAMCONFIGURATION =
f90 = gfortran
export f90
F90OPTFLAGS = -O3 -Wconversion  -fcheck=all
F90OPTFLAGS = -O3
export F90OPTFLAGS
F90USEFULFLAGS = -cpp -std=f2003
F90USEFULFLAGS = -cpp
export F90USEFULFLAGS
FL = -c ${F90OPTFLAGS} ${PROGRAMCONFIGURATION}
export FL
DIR = ${CURDIR}
export DIR
```

```
Libs = ${DIR}/Libraries/
export Libs
LIB_BLAS_LAPACK = -llapack -lblas
export LIB_BLAS_LAPACK

all: lib ana program

lib:
        cd Libraries && $(MAKE)
ana:
        cd Analysis && $(MAKE)
program:
        cd Prog && $(MAKE)

clean: cleanall
cleanall: cleanprog cleanlib cleanana
cleanprog:
        cd Prog && $(MAKE) clean
cleanlib:
        cd Libraries && $(MAKE) clean
cleanana:
        cd Analysis && $(MAKE) clean
help:
        @echo "The following are some of the valid targets of this Makefile"
        @echo "all, program, lib, ana, clean, cleanall, cleanprog, cleanlib,
            cleanana"
```

In the above, the GNU Fortan compiler `gfortran` is set.[6] We provide a set of options for compilation of the QMC code. The present options are –DMPI, –DQRREF, –DSTAB1, and –DSTAB2. They can be included in the string variable `PROGRAMCONFIGURATION` by the user, as shown above. The program can be compiled and ran either in single-thread mode (default) or in multi-threading mode (define –DMPI) using the MPI standard for parallelization. The remaining three compiler options select a particular stabilization scheme for the matrix multiplications (see Sec. 3.3.3). To compile the libraries, the analysis routines and the QMC program at once, just execute the single command:

`make`

To clean up all directories and remove the object files and executables, execute the command `make clean`. As can be seen in the above makefile, there exist also rules to compile/clean up the library, the analysis routines and the QMC program separately.

### 3.6.2  Starting a simulation

To start a simulation from scratch, the following files have to be present: `parameters` and `seeds`. To run a single-thread simulation, for example by using the parameters of one of the Hubbard models described in Sec. 4, issue the command

`./Prog/Examples.out`

To restart the code using an existing simulation as a starting point, first run the script `out_to_in.sh` to set the input configuration files.

---

[6]A known issue with the alternative Intel Fortran compiler `ifort` is the handling of automatic, temporary arrays which `ifort` allocates on the stack. For large system sizes and/or low temperatures this may lead to a runtime error. One solution is to demand allocation of arrays above a certain size on the heap instead of the stack. This is accomplished by the `ifort` compiler flag `-heap-arrays [n]` where `[n]` is the minimal size (in kilobytes, for example n=1024) of arrays that are allocated on the heap.

### 3.6.3 Error analysis

Note that the error analysis script requires the presence of the environment variable `DIR` which defines the path to the error analysis programs. So before starting the error analysis, one has to make this variable available which is done by the script `setenv.sh`. The command is

```
source ./setenv.sh
```

To perform an error analysis based on the Jackknife resampling method (Sec. 2.4.1) of the Monte Carlo bins for all observables run the script `analysis.sh` (see Sec. 3.5). In case that the parameter `N_auto` is set to a finite value the script will also trigger the computation of autocorrelation functions (Sec. 2.4.2).

## 4 Examples

### 4.1 The $SU(2)$-Hubbard model on a square lattice

To implement a Hamiltonian, the user has to provide a module which specifies the lattice, the model, as well as the observables they wish to compute. In this section, we describe the module `Hamiltonian_Examples.f90` which contains an implementation of the Hubbard model on the square lattice. A sample run for this model can be found in `Examples/Hubbard_SU2_Square/`. The input files are `parameters` and `seeds` (see Tab. 8). The output files are `info`, `confout`, and files with suffixes `_scal`, `_eq`, and `_tau` that contain the raw measurements (see Tab. 9).

The Hamiltonian reads

$$\mathcal{H} = \sum_{\sigma=1}^{2} \sum_{x,y=1}^{N_{\text{unit cell}}} c_{x\sigma}^{\dagger} T_{x,y} c_{y\sigma} + \frac{U}{2} \sum_{x} \left[ \sum_{\sigma=1}^{2} \left( c_{x\sigma}^{\dagger} c_{x\sigma} - 1/2 \right) \right]^{2} . \tag{69}$$

We can make contact with the general form of the Hamiltonian by setting: $N_{\text{fl}} = 1$, $N_{\text{col}} \equiv$ `N_SUN` $= 2$, $M_T = 1$, $T_{xy}^{(ks)} = T_{x,y}$, $M_V = N_{\text{unit cell}}$, $U_k = -\frac{U}{2}$, $V_{xy}^{(ks)} = \delta_{x,y}\delta_{x,k}$, $\alpha_{ks} = -\frac{1}{2}$ and $M_I = 0$.

### 4.1.1 Setting the Hamiltonian: `Ham_set`

The main program will call the subroutine `Ham_set` in the module `Hamiltonian_Hub.f90`. The latter subroutine defines the public variables

```
Type (Operator), dimension(:,:), allocatable  :: Op_V
Type (Operator), dimension(:,:), allocatable  :: Op_T
Integer, allocatable :: nsigma(:,:)
Integer              :: Ndim,  N_FL,  N_SUN,  Ltrot
```

which specify the model. The array `nsigma` contains the HS field. The routine `Ham_set` will first read the parameter file, then set the lattice, `Call Ham_latt`, set the hopping, `Call Ham_hop`, and set the interaction, `call Ham_V`. The parameters are read in from the file `parameters`, see Sec. 3.3.1.

**The lattice:** `Call Ham_latt` The choice `Lattice_type = "Square"` sets $\vec{a}_1 = (1,0)$ and $\vec{a}_2 = (0,1)$ and for an $L_1 \times L_2$ lattice $\vec{L}_1 = L_1\vec{a}_1$ and $\vec{L}_2 = L_2\vec{a}_2$. The call to `Call Make_Lattice( L1, L2, a1, a2, Latt)` will generate the lattice `Latt` of type `Lattice`. For the Hubbard model on the square lattice, the number of orbitals per unit cell is given by `NORB=1` such that $N_{\text{dim}} \equiv N_{\text{unit cell}} \cdot$ `NORB` $=$ `Latt%N` $\cdot$ `NORB`, since $N_{\text{unit cell}} =$ `Latt%N`.

**The hopping term:** `Call Ham_hop`  The hopping matrix is implemented as follows. We allocate an array of dimension $1 \times 1$ of type operator called `Op_T` and set the dimension for the hopping matrix to $N = N_{\text{dim}}$. One allocates and initializes this type by a single call to the subroutine `Op_make`:

```
call Op_make(Op_T(1,1),Ndim)
```

Since the hopping does not break down into small blocks, we have $\boldsymbol{P} = \mathbb{1}$ and

```
Do i= 1,Ndim
  Op_T(1,1)%P(i) = i
Enddo
```

We set the hopping matrix with

```
DO I = 1, Latt%N
   Ix = Latt%nnlist(I,1,0)
   Iy = Latt%nnlist(I,0,1)
   Op_T(1,1)%O(I  ,Ix) = cmplx(-Ham_T,    0.d0,kind(0.D0))
   Op_T(1,1)%O(Ix, I ) = cmplx(-Ham_T,    0.d0,kind(0.D0))
   Op_T(1,1)%O(I  ,Iy) = cmplx(-Ham_T,    0.d0,kind(0.D0))
   Op_T(1,1)%O(Iy, I ) = cmplx(-Ham_T,    0.d0,kind(0.D0))
   Op_T(1,1)%O(I  ,I ) = cmplx(-Ham_chem,0.d0,kind(0.D0))
ENDDO
```

Here, the integer function  `j= Latt%nnlist(I,n,m)` is defined in the lattice module and returns the index of the lattice site $\vec{I} + n\vec{a}_1 + m\vec{a}_2$. Note that periodic boundary conditions are already taken into account. The hopping parameter `Ham_T` as well as the chemical potential `Ham_chem` are read from the parameter file. To completely define the hopping we further set: `Op_T(1,1)%g = -Dtau` , `Op_T(1,1)%alpha = cmplx(0.d0,0.d0,` `kind(0.D0))` and call the routine `Op_set(Op_T(1,1))` so as to generate the unitary transformation and eigenvalues as specified in Table 3. Recall that for the hopping, the variable `Op_set(Op_T(1,1))%type` is not required. Note that although a checkerboard decomposition is not used here, it can be implemented by considering a larger number of sparse hopping matrices.

**The interaction term:** `Call Ham_V`  To implement this interaction, we allocate an array of `Operator` type. The array is called `Op_V` and has dimensions $N_{\text{dim}} \times N_{\text{fl}} = N_{\text{dim}} \times 1$. We set the dimension for the interaction term to $N = 1$, and allocate and initialize this array of type `Operator` by repeatedly calling the subroutine `Op_make`:

```
do i  = 1,Ndim
   call Op_make(Op_V(i,1),1)
enddo
```

For each lattice site $i$, the matrices $\boldsymbol{P}$ are of dimension $1 \times N_{\text{dim}}$ and have only one non-vanishing entry. Thereby we can set:

```
Do i = 1,Ndim
   Op_V(i,1)%P(1)    = i
```

```
   Op_V(i,1)%O(1,1) = cmplx(1.d0,0.d0, kind(0.D0))
   Op_V(i,1)%g      = sqrt(cmplx(-dtau*ham_U/dble(N_SUN),0.D0,kind(0.D0)))
   Op_V(i,1)%alpha  = cmplx(-0.5d0,0.d0, kind(0.D0))
   Op_V(i,1)%type   = 2
   Call Op_set( Op_V(i,1) )
Enddo
```

so as to completely define the interaction term.

### 4.1.2 Observables

At this point, all the information for the simulation to start has been provided. The code will sequentially go through the operator list `Op_V` and update the fields. Between time slices `LOBS_ST` and `LOBS_EN` the main program will call the routine `Obser(GR,Phase,Ntau)` which is provided by the user and handles equal time correlation functions. If `Ltau=1` the main program will call the routine `ObserT(NT, GT0,G0T,G00,GTT, PHASE)` which is again provided by the user and handles imaginary-time displaced correlation functions.

The user will have to implement the observables he/she wants to compute. Here we will describe how to proceed.

**Allocating space for the observables:** `Call Alloc_obs(Ltau)` For four scalar or vector observables, the user will have to declare the following:

```
Allocate ( Obs_scal(4) )
Do I = 1,Size(Obs_scal,1)
   select case (I)
   case (1)
      N = 2;  Filename ="Kin"
   case (2)
      N = 1;  Filename ="Pot"
   case (3)
      N = 1;  Filename ="Part"
   case (4)
      N = 1,  Filename ="Ener"
   case default
      Write(6,*) ' Error in Alloc_obs '
   end select
   Call Obser_Vec_make(Obs_scal(I),N,Filename)
enddo
```

Here, `Obs_scal(1)` contains a vector of two observables so as to account for the $x$- and $y$-components of the kinetic energy for example.

For equal time correlation functions we allocate `Obs_eq` of type `Obser_Latt`. Here we include the calculation of spin-spin and density-density correlation functions alongside equal time Green functions.

```
Allocate ( Obs_eq(4) )
Do I = 1,Size(Obs_eq,1)
   select case (I)
   case (1)
      Ns = Latt%N; No = Norb;  Filename ="Green"
```

```
   case (2)
      Ns = Latt%N; No = Norb;  Filename ="SpinZ"
   case (3)
      Ns = Latt%N; No = Norb;  Filename ="SpinXY"
   case (4)
      Ns = Latt%N; No = Norb;  Filename ="Den"
   case default
      Write(6,*) ' Error in Alloc_obs '
   end select
   Nt = 1
   Call Obser_Latt_make(Obs_eq(I),Ns,Nt,No,Filename)
enddo
```

For the Hubbard model `Norb = 1` and for equal time correlation functions `Nt = 1`. If `Ltau = 1` then the code will allocate space for time displaced quantities. The same structure as for equal time correlation functions will be used albeit with `Nt = Ltrot + 1`. At the beginning of each bin, the main program will set the bin observables to zero by calling the routine `Init_obs(Ltau)`. The user does not have to edit this routine.

**Measuring equal time observables:** `Obser(GR,Phase,Ntau)`   The equal time Green function,

$$\texttt{GR(x,y,}\sigma\texttt{)} = \langle c_{x,\sigma} c^{\dagger}_{y,\sigma} \rangle, \tag{70}$$

the phase factor `phase` [Eq. (64)], and time slice `Ntau` are provided by the main program. Here, $x$ and $y$ label both unit cell as well as the orbital within the unit cell. For the Hubbard model described here, $x$ corresponds to the unit cell. The Green function does not depend on the color index, and is diagonal in flavor. For the $SU(2)$ symmetric implementation there is only one flavor, $\sigma = 1$ and the Green function is independent on the spin index. This renders the calculation of the observables particularly easy.

An explicit calculation of the potential energy $\langle U \sum_{\vec{i}} \hat{n}_{\vec{i},\uparrow} \hat{n}_{\vec{i},\downarrow} \rangle$ reads

```
Obs_scal(2)%N        = Obs_scal(2)%N + 1
Obs_scal(2)%Ave_sign = Obs_scal(2)%Ave_sign + Real(ZS,kind(0.d0))
Do i = 1,Ndim
   Obs_scal(2)%Obs_vec(1)=Obs_scal(2)%Obs_vec(1)+(1-GR(i,i,1))**2*Ham_U*ZS*ZP
Enddo
```

Here ZS = $\text{sign}(C)$ [see Eq. (23)], ZP $= \frac{e^{-S(C)}}{\Re\left[e^{-S(C)}\right]}$ [see Eq. (64)] and `Ham_U` corresponds to the Hubbard-$U$ term.

Equal time correlations are also computed in this routine. As an explicit example, we consider the equal time density-density correlation:

$$\langle n_{\vec{i},\alpha} n_{\vec{j},\beta} \rangle - \langle n_{\vec{i},\alpha} \rangle \langle n_{\vec{j},\beta} \rangle \, . \tag{71}$$

For the calculation of such quantities, it is convenient to define:

$$\texttt{GRC(x,y,s)} = \delta_{x,y} - \texttt{GR(y,x,s)} \quad , \tag{72}$$

such that `GRC(x,y,s)` corresponds to $\langle \langle \hat{c}^{\dagger}_{x,s} \hat{c}_{y,s} \rangle \rangle$. In the program code, the calculation of the equal time density-density correlation function looks as follows:

```
Obs_eq(4)%N      = Obs_eq(4)%N + 1  ! Even if it is redundant, each observable
                                    ! carries its own counter and sign.
Obs_eq(4)%Ave_sign = Obs_eq(4)%Ave_sign + Real(ZS,kind(0.d0))
Do I1 = 1,Ndim
   I   = List(I1,1)                 ! = I1  (The Hubbard model  on the square
   no_I = List(I1,2)                ! = 1   lattice has one orbital per unit
                                    !       cell)
   Do J1 = 1,Ndim
      J   = List(J1,1)
      no_J = List(J1,2)
      imj = latt%imj(I,J)
      Obs_eq(4)%Obs_Latt(imj,1,no_I,no_J) = &
            & Obs_eq(4)%Obs_Latt(imj,1,no_I,no_J) + &
            & (    GRC(I1,I1,1) * GRC(J1,J1,1) * N_SUN * N_SUN     + &
            &      GRC(I1,J1,1) * GR(I1,J1,1) * N_SUN   ) * ZP * ZS
   Enddo
   Obs_eq(4)%Obs_Latt0(no_I) = &
   & Obs_eq(4)%Obs_Latt0(no_I)+GRC(I1,I1,1) * N_SUN * ZP * ZS
Enddo
```

Note that we consider the square lattice of the single site Hubbard model as a special case of a multiorbital problem as described in Sec. 4.3.1 At the end of each bin the main program will call the routine `Pr_obs(LTAU)`. This routine will append the result of the bins in the specified file, with appropriate suffix.

**Measuring time displaced observables:** `ObserT(NT, GT0,G0T,G00,GTT, PHASE)`   This subroutine is called by the main program at the beginning of each sweep, provided that `LTAU` is set to unity. `NT` runs from 0 to `Ltrot` and denotes the imaginary time difference. For a given time displacement, the main program provides:

$$
\begin{aligned}
\texttt{GT0(x,y,s)} &= \langle\langle \hat{c}_{x,s}(Nt\Delta\tau)\hat{c}^{\dagger}_{y,s}(0)\rangle\rangle = \langle\langle \mathcal{T}\hat{c}_{x,s}(Nt\Delta\tau)\hat{c}^{\dagger}_{y,s}(0)\rangle\rangle \\
\texttt{G0T(x,y,s)} &= -\langle\langle \hat{c}^{\dagger}_{y,s}(Nt\Delta\tau)\hat{c}_{x,s}(0)\rangle\rangle = \langle\langle \mathcal{T}\hat{c}_{x,s}(0)\hat{c}^{\dagger}_{y,s}(Nt\Delta\tau)\rangle\rangle \\
\texttt{G00(x,y,s)} &= \langle\langle \hat{c}_{x,s}(0)\hat{c}^{\dagger}_{y,s}(0)\rangle\rangle \\
\texttt{GTT(x,y,s)} &= \langle\langle \hat{c}_{x,s}(Nt\Delta\tau)\hat{c}^{\dagger}_{y,s}(Nt\Delta\tau)\rangle\rangle
\end{aligned}
\tag{73}
$$

In the above we have omitted the color index since the Green functions are color independent. The time displaced spin-spin correlations $4\langle\langle \hat{S}^z_{\vec{i}}(\tau)\hat{S}^z_{\vec{j}}(0)\rangle\rangle$ are thereby given by:

$$
4\langle\langle \hat{S}^z_{\vec{i}}(\tau)\hat{S}^z_{\vec{j}}(0)\rangle\rangle = -2\,\texttt{G0T(J1,I1,1)}\,\texttt{GT0(I1,J1,1)} \ .
\tag{74}
$$

Note that the above holds for the $SU(2)$ HS transformation discussed in this chapter. The handling of time displaced correlation functions is identical to that of equal time correlations.

### 4.1.3   Numerical precision

The directory `Examples/Hubbard_SU2_Square` contains an example simulation of the $4 \times 4$ Hubbard model at $U/t = 4$ and $\beta t = 10$. Information on the numerical stability is included in the following lines of the corresponding file `info`:

```
Precision Green  Mean, Max : 1.2918865817224671E-014  4.0983018995027644E-011
Precision Phase, Max     : 5.0272908791449966E-012
Precision tau    Mean, Max : 8.4596701790588625E-015  3.5033530012121281E-011
```

showing the mean and maximum difference between the *wrapped* and from scratched computed equal and time displaced Green functions [7]. A stable code should produce results where the mean difference is smaller than the stochastic error. The above example shows a very stable simulation since the Green function is of order one.

## 4.2 The $M_z$-Hubbard model on a square lattice

The Hubbard Hamiltonian can equally be written as:

$$\mathcal{H} = \sum_{\sigma=1}^{2} \sum_{x,y=1}^{N_{\text{unit cell}}} c_{x\sigma}^{\dagger} T_{x,y} c_{y\sigma} - \frac{U}{2} \sum_{x} \left[ c_{x,\uparrow}^{\dagger} c_{x\uparrow} - c_{x,\downarrow}^{\dagger} c_{x\downarrow} \right]^{2} . \tag{75}$$

We can make contact with the general form of the Hamiltonian (see Eq. 2) by setting: $N_{\text{fl}} = 2$, $N_{\text{col}} \equiv$ N_SUN $= 1$, $M_T = 1$, $T_{xy}^{(ks)} = T_{x,y}$, $M_V = N_{\text{unit cell}}$, $U_k = \frac{U}{2}$, $V_{xy}^{(k,s=1)} = \delta_{x,y}\delta_{x,k}$, $V_{xy}^{(k,s=2)} = -\delta_{x,y}\delta_{x,k}$, $\alpha_{ks} = 0$ and $M_I = 0$. The coupling of the HS fields to the $z$-component of the magnetization breaks the $SU(2)$ spin symmetry. Nevertheless the $z$-component of the spin remains a good quantum number such that the imaginary-time propagator – for a given HS field – is block diagonal in this quantum number. This corresponds to the flavor index which runs from one to two labelling spin up and spin down degrees of freedom. In the parameter file listed in Sec. 3.3.1 setting the model variable to Hubbard_Mz will carry out the simulation in the above representation. With respect to the $SU(2)$ case, the changes required in the Hamiltonian_Examples.f90 module are minimal and essentially effect only the interaction term, and the calculation of observables. We note that in this formulation the hopping matrix can be flavor dependent such that a Zeeman magnetic field can be introduced. If the chemical potential is set to zero, this will not generate a negative sign problem [37, 88, 89]. A sample run for this model can be found in Examples/Hubbard_Mz_Square/. The input files are parameters and seeds (see Tab. 8). The output files are info, confout, and files with suffixes _scal, _eq, and _tau that contain the raw measurements (see Tab. 9).

### 4.2.1 The interaction term: Call Ham_V

The interaction term is now given by:

```fortran
Allocate(Op_V(Ndim,N_FL))
do nf = 1,N_FL
   do i  = 1, Ndim
      Call Op_make(Op_V(i,nf),1)
   enddo
enddo
Do nf = 1,N_FL
   nc = 0
   X = 1.d0
   if (nf == 2) X = -1.d0
   Do i = 1,Ndim
      nc = nc + 1
      Op_V(nc,nf)%P(1) = I
      Op_V(nc,nf)%O(1,1) = cmplx(1.d0, 0.d0, kind(0.D0))
      Op_V(nc,nf)%g       = X*SQRT(CMPLX(DTAU*ham_U/2.d0, 0.D0, kind(0.D0)))
      Op_V(nc,nf)%alpha   = cmplx(0.d0, 0.d0, kind(0.D0))
      Op_V(nc,nf)%type    = 2
      Call Op_set( Op_V(nc,nf) )
   Enddo
Enddo
```

In the above, one will see explicitly that there is a sign difference between the coupling of the HS field in the two flavor sectors.

### 4.2.2   The measurements: `Call Obser, Call ObserT`

Since the spin up and spin down Green functions differ for a given HS configuration, the Wick decomposition will take a different form. In particular, the equal time spin-spin correlation functions $4\langle\langle\hat{S}^z_i\hat{S}^z_j\rangle\rangle$ calculated in the subroutine `Obser` will take the form:

$$4\langle\langle\hat{S}^z_x\hat{S}^z_y\rangle\rangle = \texttt{GRC(x,y,1) * GR(x,y,1) + GRC(x,y,2) * GR(x,y,2) +}$$
$$\texttt{(GRC(x,x,2) - GRC(x,x,1))*(GRC(y,y,2) - GRC(y,y,1))}$$

Here, GRC is defined in Eq. 72. Equivalent changes will have to be carried out for other equal time and time displaced observables.

Apart from these modifications, the program will run in exactly the same manner as for the $SU(2)$ case.

### 4.2.3   Numerical precision

The directory `Examples/Hubbard_Mz_Square` contains an example simulation of the $4 \times 4$ Hubbard model at $U/t = 4$ and $\beta t = 10$. Information on the numerical stability is included in the following lines of the corresponding file `info`:

```
Precision Green  Mean, Max : 5.0823874429126405E-011  5.8621144596315844E-006
Precision Phase, Max       : 0.0000000000000000
Precision tau    Mean, Max : 1.5929357848647394E-011  1.0985132530727526E-005
```

This is still an excellent precision but nevertheless choosing a HS field which couples to the z-component of the magnetization apparently leads to numerical results that are a couple of order of magnitudes less precise than a HS decomposition coupling to the charge (compare with Sec. 4.1.3).

## 4.3   The $SU(2)$-Hubbard model on the honeycomb lattice

The Hamilton module `Hamiltonian_Examples.f90` can also carry out simulations for the Hubbard model on the Honeycomb lattice by setting in the parameter file `Lattice_type= "Honeycomb"` (see Sec. 3.3.1). A sample run for this model can be found in `Examples/ Hubbard_SU2_Honeycomb/`. The input files are `parameters` and `seeds` (see Tab. 8). The output files are `info`, `confout`, and files with suffixes `_scal`, `_eq`, and `_tau` that contain the raw measurements (see Tab. 9).

### 4.3.1   Working with multi-orbital unit cells: `Call Ham_Latt`

This model is an example of a multi-orbital unit cell, and the aim of this section is to document how to implement this in the code. The Honeycomb lattice is a triangular Bravais lattice with two orbitals per unit cell. The routine `Ham_Latt` will set:

```
Norb    = 2
N_coord = 3
a1_p(1) =  1.D0   ; a1_p(2) =  0.d0
a2_p(1) =  0.5D0  ; a2_p(2) =  sqrt(3.D0)/2.D0
L1_p    =  dble(L1) * a1_p
L2_p    =  dble(L2) * a2_p
```

and then call `Make_Lattice(L1_p,L2_p,a1_p,a2_p,Latt)` so as to generate the triangular lattice. The coordination number of this lattice is  `N_coord=3`  and the number

of orbitals per unit cell corresponds to NORB=2. The total number of orbitals is thereby: $N_{dim}$=Latt%N*NORB. To easily keep track of the orbital and unit cell, we define a super-index as shown below:

```
Allocate (List(Ndim,2), Invlist(Latt%N,Norb))
nc = 0
Do I = 1,Latt%N                             ! Unit-cell index
  Do no = 1,Norb                            ! Orbital index
    nc = nc + 1                             ! Super-index labeling unit cell
                                            ! and orbital
    List(nc,1) = I                          ! Unit-cell  of super index  nc
    List(nc,2) = no                         ! Orbital of super index nc
    Invlist(I,no) = nc                      ! Super-index for given  unit cell
                                            ! and orbital

  Enddo
Enddo
```

With the above lists one can run through all the orbitals and at each time keep track of the unit-cell and orbital index. We note that when translation symmetry is completely absent one can work with a single unit cell, and the number of orbitals will then correspond to the number of lattice sites.

### 4.3.2  The hopping term: `Call Ham_Hop`

Some care has to be taken when setting the hopping matrix. In the Hamilton module `Hamiltonian_Examples.f90` we do this in the following way:

```
DO I = 1, Latt%N                            ! Loop over unit cell
  do no = 1,Norb                            ! Runs over orbitals and
                                            ! sets the chemical potential
    I1 = invlist(I,no)
    Op_T(nc,n)%O(I1 ,I1) = cmplx(-Ham_chem, 0.d0, kind(0.D0))
  enddo
  I1 = Invlist(I,1)                         ! Orbital A of unit cell I
  Do nc1 = 1,N_coord                        ! Loop over coordination  number
    select case (nc1)
    case (1)
       J1 = invlist(I,2)                    ! Orbital B of unit cell i
    case (2)
       J1=invlist(Latt%nnlist(I,1,-1),2)    ! Orbital B of unit cell i+a_1-a_2
    case (3)
       J1=invlist(Latt%nnlist(I,0,-1),2)    ! Orbital B of unit cell i-a_2
    case default
       Write(6,*) ' Error in  Ham_Hop '
    end select
    Op_T(nc,n)%O(I1,J1) = cmplx(-Ham_T,    0.d0, kind(0.D0))
    Op_T(nc,n)%O(J1,I1) = cmplx(-Ham_T,    0.d0, kind(0.D0))
  Enddo
Enddo
```

As apparent from the above, hopping matrix elements are non-zero only between the *A* and *B* sublattices.

### 4.3.3  Observables: `Call Obser, Call ObserT`

In the multi-orbital case, the correlation functions have additional orbital indices. This is automatically taken care of in the routines `Call Obser` and `Call ObserT` since we have

already considered the Hubbard model on the square lattice to correspond to a multi-orbital unit cell albeit with the special choice of one orbital per unit cell.

## 4.4 The $SU(2)$-Hubbard model on a square lattice coupled to a transverse Ising field

The model we consider here is very similar to the above, but has an additional coupling to a transverse field:

$$\mathcal{H} = \sum_{\sigma=1}^{2}\sum_{x,y} c_{x\sigma}^{\dagger} T_{x,y} c_{y\sigma} + \frac{U}{2}\sum_{x}\left[\sum_{\sigma=1}^{2}\left(c_{x\sigma}^{\dagger}c_{x\sigma}-1/2\right)\right]^{2} + \xi\sum_{\sigma,\langle x,y\rangle}\hat{Z}_{\langle x,y\rangle}\left(c_{x\sigma}^{\dagger}c_{y\sigma}+h.c.\right)$$
$$-h\sum_{\langle x,y\rangle}\hat{X}_{\langle x,y\rangle} - J\sum_{\langle\langle x,y\rangle\langle x',y'\rangle\rangle}\hat{Z}_{\langle x,y\rangle}\hat{Z}_{\langle x',y'\rangle} \quad (76)$$

We can make contact with the general form of the Hamiltonian by setting: $N_{\mathrm{fl}}=1$, $N_{\mathrm{col}}\equiv\mathtt{N\_SUN}=2$, $M_{T}=1$, $T_{xy}^{(ks)}=T_{x,y}$, $M_{V}=N_{\mathrm{unit\ cell}}\equiv N_{\mathrm{dim}}$, $U_{k}=-\frac{U}{2}$, $V_{xy}^{(ks)}=\delta_{x,y}\delta_{x,k}$, $\alpha_{ks}=-\frac{1}{2}$ and $M_{I}=2N_{\mathrm{unit\ cell}}$. The last two terms of the above Hamiltonian describe a transverse Ising field model on the bonds of the square lattice. This type of Hamiltonian has recently been extensively discussed [19, 22, 90]. Here we adopt the notation of Ref. [19]. Note that $\langle\langle x,y\rangle\langle x',y'\rangle\rangle$ denotes nearest neighbor bonds. The modifications required to generalize the Hubbard model code to the above model are two-fold. First, one has to specify the function `Real(Kind=8)functionS0(n,nt)`, and second, modify the interaction `Call Ham_V`. A sample run for this model can be found in `Examples/Hubbard_SU2_Ising_Square/`. The input files are `parameters` and `seeds` (see Tab. 8). The output files are `info`, `confout`, and files with suffixes `_scal`, `_eq`, and `_tau` that contain the raw measurements (see Tab. 9).

### 4.4.1 The Ising term

Since the Ising field lives on bonds we have to provide a data structure defining this quantity. A bond has an anchor site as well as an orientation. The routine `Setup_Ising_action` initializes the arrays `L_bond` and `L_bond_inv` that contain this information.

```
nc = 0
Do n_orientation = 1,N_coord
Do I = 1, Latt%N
   nc = nc + 1
   L_bond(I,n_orientation) = nc
   L_bond_inv(nc,1) = I
   L_bond_inv(nc,2) = n_orientation
enddo
enddo
```

The two legs of the bond are given by the anchor $\vec{I}$ and $\vec{I}+\vec{a}_{n_{\mathrm{orientation}}}$.

### 4.4.2 The interaction term: `Call Ham_V`

The dimension of `Op_V` is now $(M_{I}+M_{V})\times N_{\mathrm{fl}}=((N_{coord}+1)N_{\mathrm{dim}})\times 1$ since each site has $N_{coord}=2$ bonds for the square lattice.

```
do i  = 1,N_coord*Ndim                      ! Runs over bonds for Ising inter.
  call Op_make(Op_V(i,1),2)
```

```
enddo
do i  =  N_coord*Ndim+1, (N_coord+1)*Ndim ! Runs over sites for Hubbard inter.
  call Op_make(Op_V(i,1),1)
enddo
```

The first `N_coord*Ndim` operators run through the $2N$ bonds of the square lattice and are given by:

```
Do nc = 1,Ndim*N_coord              ! Runs over bonds. Coordination number = 2.
                                    ! For the square lattice Ndim = Latt%N

  I1 = L_bond_inv(nc,1)             ! Anchor of the bond
                                    ! L_bond_inv is setup in Setup_Ising_action
  if ( L_bond_inv(nc,2)  == 1 ) I2 = Latt%nnlist(I1,1,0) ! Second site of
  if ( L_bond_inv(nc,2)  == 2 ) I2 = Latt%nnlist(I1,0,1) ! the bond
  Op_V(nc,1)%P(1) = I1
  Op_V(nc,1)%P(2) = I2
  Op_V(nc,1)%O(1,2) = cmplx(1.d0 ,0.d0, kind(0.D0))
  Op_V(nc,1)%O(2,1) = cmplx(1.d0 ,0.d0, kind(0.D0))
  Op_V(nc,1)%g      = cmplx(-dtau*Ham_xi,0.D0,kind(0.D0))
  Op_V(nc,1)%alpha  = cmplx(0d0,0.d0, kind(0.D0))
  Op_V(nc,1)%type   = 1
Enddo
```

Here, `ham_xi` defines the coupling strength between the Ising and fermion degree of freedom. As for the Hubbard case, the last `Ndim` operators read:

```
nc = N_coord*Ndim
Do i = 1, Ndim
    nc = nc + 1
    Op_V(nc,1)%P(1)  = i
    Op_V(nc,1)%O(1,1)= cmplx(1.d0  ,0.d0, kind(0.D0))
    Op_V(nc,1)%g     = sqrt(cmplx(-dtau*ham_U/(DBLE(N_SUN)),0.D0,kind(0.D0)))
    Op_V(nc,1)%alpha = cmplx(-0.5d0,0.d0, kind(0.D0))
    Op_V(nc,1)%type  = 2
Enddo
```

### 4.4.3  The function `Real (Kind=8) function S0(n,nt)`

As mentioned above, a configuration now includes both HS spins and Ising spins and is given by

$$C = \left\{ s_{i,\tau}, l_{j,\tau} \text{ with } i = 1 \cdots M_I, j = 1 \cdots M_V, \tau = 1, L_{Trotter} \right\} . \tag{77}$$

This configuration is stored in the integer array `nsigma(M_V + M_I, Ltrot)`. With the above ordering of Hubbard and Ising interaction terms, and a for a given imaginary time, the first `2*Ndim` fields correspond to the Ising interaction and the next `Ndim` ones to the Hubbard interaction. The first argument of the function S0, namely `n`, corresponds to the index of the operator string `Op_V(n,1)`. If `Op_V(n,1)%type = 2` then `S0(n,nt)` returns 1. Note that `type=2` refers to spins that stem from a HS transformation. If however `Op_V(n,1)%type = 1` then function S0 returns

$$\frac{e^{-S_{0,I}\left(s_{1,\tau},\cdots,-s_{n,\tau},\cdots s_{M_I,\tau}\right)}}{e^{-S_{0,I}\left(s_{1,\tau},\cdots,s_{n,\tau},\cdots s_{M_I,\tau}\right)}} \tag{78}$$

That is, if $n \leq 2*$`Ndim`, `S0(n,nt)` returns the ratio of the new weight to the old weight of the Ising Hamiltonian upon flipping a single Ising spin $s_{n,\tau}$. Otherwise, `S0(n,nt)` returns unity.

# 5 Miscellaneous

## 5.1 Other models

The aim of this section is to briefly mention a small selection of other models that can be studied using the QMC code of the ALF project.

### 5.1.1 Kondo lattice model

Simulating the Kondo lattice with the QMC code of the ALF project requires rewriting of the model along the lines of Refs. [20, 21, 91]. Adopting the notation of these articles, the Hamiltonian that one will simulate reads:

$$\hat{\mathscr{H}} = \underbrace{-t \sum_{\langle \vec{i}, \vec{j} \rangle, \sigma} \left( \hat{c}_{\vec{i},\sigma}^{\dagger} \hat{c}_{\vec{j},\sigma} + \text{H.c.} \right)}_{\equiv \hat{\mathscr{H}}_t} - \frac{J}{4} \sum_{\vec{i}} \left( \sum_{\sigma} \hat{c}_{\vec{i},\sigma}^{\dagger} \hat{f}_{\vec{i},\sigma} + \hat{f}_{\vec{i},\sigma}^{\dagger} \hat{c}_{\vec{i},\sigma} \right)^2 + \underbrace{\frac{U}{2} \sum_{\vec{i}} \left( \hat{n}_{\vec{i}}^f - 1 \right)^2}_{\equiv \hat{\mathscr{H}}_U}. \tag{79}$$

This form is included in the general Hamiltonian (2) such that the above Hamiltonian can be implemented in our program package. The relation to the Kondo lattice model follows from expanding the square of the hybridization to obtain:

$$\hat{\mathscr{H}} = \hat{\mathscr{H}}_t + J \sum_{\vec{i}} \left( \hat{\vec{S}}_{\vec{i}}^c \cdot \hat{\vec{S}}_{\vec{i}}^f + \hat{\eta}_{\vec{i}}^{z,c} \cdot \hat{\eta}_{\vec{i}}^{z,f} - \hat{\eta}_{\vec{i}}^{x,c} \cdot \hat{\eta}_{\vec{i}}^{x,f} - \hat{\eta}_{\vec{i}}^{y,c} \cdot \hat{\eta}_{\vec{i}}^{y,f} \right) + \hat{\mathscr{H}}_U. \tag{80}$$

where the $\eta$-operators relate to the spin-operators via a particle-hole transformation in one spin sector:

$$\hat{\eta}_{\vec{i}}^{\alpha} = \hat{P}^{-1} \hat{S}_{\vec{i}}^{\alpha} \hat{P} \quad \text{with} \quad \hat{P}^{-1} \hat{c}_{\vec{i},\uparrow} \hat{P} = (-1)^{i_x+i_y} \hat{c}_{\vec{i},\uparrow}^{\dagger} \quad \text{and} \quad \hat{P}^{-1} \hat{c}_{\vec{i},\downarrow} \hat{P} = \hat{c}_{\vec{i},\downarrow} \tag{81}$$

Since the $\hat{\eta}^f$- and $\hat{S}^f$-operators do not alter the parity $[(-1)^{\hat{n}_{\vec{i}}^f}]$ of the $f$-sites,

$$\left[ \hat{\mathscr{H}}, \hat{\mathscr{H}}_U \right] = 0. \tag{82}$$

Thereby, and for positive values of $U$, doubly occupied or empty $f$-sites – corresponding to even parity sites – are suppressed by a Boltzmann factor $e^{-\beta U/2}$ in comparison to odd parity sites. Choosing $\beta U$ adequately essentially allows to restrict the Hilbert space to odd parity $f$-sites. In this Hilbert space $\hat{\eta}^{x,f} = \hat{\eta}^{y,f} = \hat{\eta}^{z,f} = 0$ such that the Hamiltonian (79) reduces to the Kondo lattice model.

### 5.1.2 $SU(N)$-Hubbard-Heisenberg models

$SU(2N)$-Hubbard-Heisenberg [26, 27] models can be written as:

$$\hat{\mathscr{H}} = \underbrace{-t \sum_{\langle \vec{i}, \vec{j} \rangle} \left( \vec{\hat{c}}_{\vec{i}}^{\dagger} \vec{\hat{c}}_{\vec{j}} + \text{H.c.} \right)}_{\equiv \hat{\mathscr{H}}_t} \underbrace{- \frac{J}{2N} \sum_{\langle \vec{i}, \vec{j} \rangle} \left( \hat{D}_{\vec{i},\vec{j}}^{\dagger} \hat{D}_{\vec{i},\vec{j}} + \hat{D}_{\vec{i},\vec{j}} \hat{D}_{\vec{i},\vec{j}}^{\dagger} \right)}_{\equiv \hat{\mathscr{H}}_J} + \underbrace{\frac{U}{N} \sum_{\vec{i}} \left( \vec{\hat{c}}_{\vec{i}}^{\dagger} \vec{\hat{c}}_{\vec{i}} - \frac{N}{2} \right)^2}_{\equiv \hat{\mathscr{H}}_U} \tag{83}$$

Here, $\vec{\hat{c}}_{\vec{i}}^{\dagger} = (\hat{c}_{\vec{i},1}^{\dagger}, \hat{c}_{\vec{i},2}^{\dagger}, \cdots, \hat{c}_{\vec{i},N}^{\dagger})$ is an $N$-flavored spinor, and $\hat{D}_{\vec{i},\vec{j}} = \vec{\hat{c}}_{\vec{i}}^{\dagger} \vec{\hat{c}}_{\vec{j}}$. To use the QMC code of the ALF package to simulate this model, one will rewrite the $J$-term as a sum of perfect squares,

$$\hat{\mathscr{H}}_J = -\frac{J}{4N} \sum_{\langle \vec{i}, \vec{j} \rangle} \left( \hat{D}_{\langle \vec{i}, \vec{j} \rangle}^{\dagger} + \hat{D}_{\langle \vec{i}, \vec{j} \rangle} \right)^2 - \left( \hat{D}_{\langle \vec{i}, \vec{j} \rangle}^{\dagger} - \hat{D}_{\langle \vec{i}, \vec{j} \rangle} \right)^2, \tag{84}$$

so to manifestly bring it into the form of the general Hamiltonian(2). It is amusing to note that setting the hopping $t = 0$, charge fluctuations will be suppressed by the Boltzmann factor $e^{-\beta U/N\left(\vec{c}_i^\dagger \vec{c}_i - \frac{N}{2}\right)^2}$ since in this case $\left[\hat{\mathcal{H}}_J, \hat{\mathcal{H}}_U\right] = 0$. This provides a route to use the auxiliary field QMC algorithm to simulate – free of the sign problem – $SU(2N)$-Heisenberg models in the self-adjoint antisymmetric representation [7]. For odd values of $N$ recent progress in our understanding of the origins of the sign problem [40] allows us to simulate a set of non-trivial Hamiltonians [19,92], without encountering the sign problem.

## 5.2 Performance, memory requirements and parallelization

As mentioned in the introduction, the auxiliary field QMC algorithm scales linearly in inverse temperature $\beta$ and cubic in the volume $N_{\text{dim}}$. Using fast updates, a single spin flip requires $(N_{\text{dim}})^2$ operations to update the Green function upon acceptance. As there are $L_{\text{Trotter}} \times N_{\text{dim}}$ spins to be visited, the total computational cost for one sweep is of the order of $\beta (N_{\text{dim}})^3$. This operation dominates the performance, see Fig. 2. A profiling analysis of our code shows that 80-90% of the CPU time is spend in ZGEMM calls of the BLAS library provided in the MKL package by Intel. Consequently, the single-core performance is next to optimal.

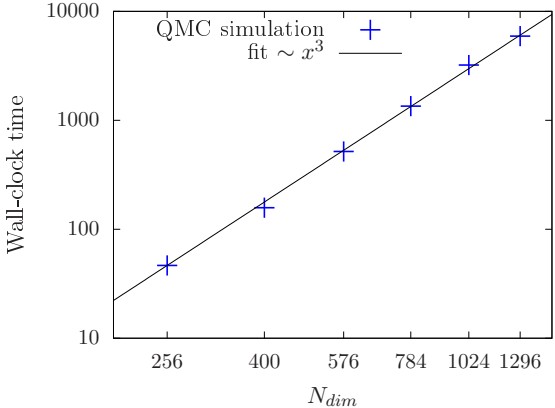

Figure 2: Volume scaling behavior of the auxiliary field QMC code of the ALF project on Super-MUC (phase 2/Haswell nodes) at the LRZ in Munich. The number of sites $N_{\text{dim}}$ corresponds to the system volume. The plot confirms that the leading scaling order is due to matrix multiplications such that the runtime is dominated by calls to ZGEMM.

For the implementation which scales linearly in $\beta$, one has to store $L_{\text{Trotter}}/\texttt{NWrap}$ intermediate propagation matrices of dimension $N \times N$. For large lattices and/or low temperatures this dominates the total memory requirements that can exceed 2 GB memory for a sequential version.

At the heart of Monte Carlo schemes lies a random walk through the given configuration space. This is easily parallalized via MPI by associating one random walker to each MPI task. For each task, we start from a random configuration and have to invest the autocorrelation time $T_{\text{auto}}$ to produce an equilibrated configuration. Additionally we can also profit from an OpenMP parallelized version of the BLAS/LAPACK library for an additional speedup, which also effects equilibration overhead $N_{\text{MPI}} \times T_{\text{auto}}/N_{\text{OMP}}$, where $N_{\text{MPI}}$ is the number of cores and $N_{\text{OMP}}$ the number of OpenMP threads. For a given number of independent measurements

---

[7] This corresponds to a Young tableau with single column and $N/2$ rows.

$N_{\text{meas}}$, we therefore need a wall-clock time given by

$$T = \frac{T_{\text{auto}}}{N_{\text{OMP}}} \left( 1 + \frac{N_{\text{meas}}}{N_{\text{MPI}}} \right). \tag{85}$$

As we typically have $N_{\text{meas}}/N_{\text{MPI}} \gg 1$, the speedup is expected to be almost perfect, in accordance with the performance test results for the auxiliary field QMC code on SuperMUC [see Fig. 3 (a)].

For many problem sizes, 2 GB memory per MPI task (random walker) suffices such that we typically start as many MPI tasks as there are physical cores per node. Due to the large amount of CPU time spent in MKL routines, we do not profit from the hyper-threading option. For large systems, the memory requirement increases and this is tackled by increasing the amount of OpenMP threads to decrease the stress on the memory system and to simultaneously reduce the equilibration overhead [see Fig. 3 (b)]. For the displayed speedup, it was crucial to pin the MPI tasks as well as the OpenMP threads in a pattern which keeps the threads as compact as possible to profit from a shared cache. This also explains the drop in efficiency from 14 to 28 threads where the OpenMP threads are spread over both sockets.

We store the field configurations of the random walker as checkpoints, such that a long simulation can be easily split into several short simulations. This procedure allows us to take advantage of chained jobs using the dependency chains provided by the batch system.

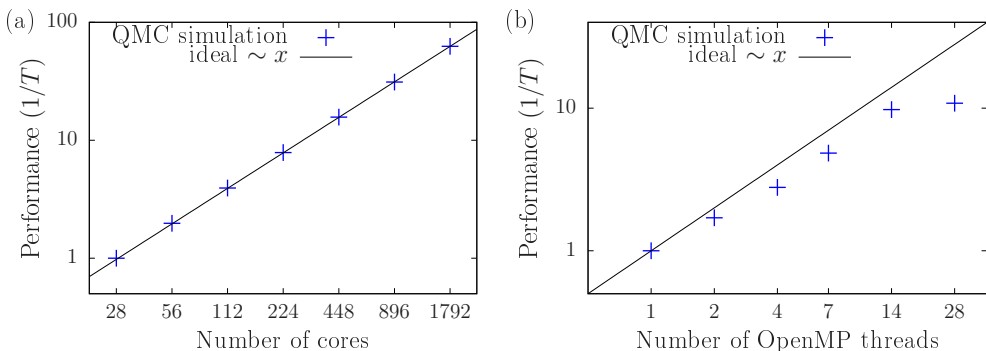

Figure 3: MPI (a) and OpenMP (b) scaling behavior of the auxiliary field QMC code of the ALF project on SuperMUC (phase 2/Haswell nodes) at the LRZ in Munich. The MPI performance data was normalized to 28 cores and was obtained using a problem size of $N_{\text{dim}} = 400$. This is a medium to small system size that is the least favorable in terms of MPI synchronization effects. The OpenMP performance data was obtained using a problem size of $N_{\text{dim}} = 1296$. Employing 2 and 4 OpenMP threads introduces some synchronization/management overhead such that the per-core performance is slightly reduced, compared to the single thread efficiency. Further increasing the amount of threads to 7 and 14 keeps the efficiency constant. The drop in performance of the 28 thread configuration is due to the architecture as the threads are now spread over both sockets of the node. To obtain the above results, it was crucial to pin the processes in a fashion that keeps the OpenMP threads as compact as possible.

## 6 Conclusions and Future Releases

In its present form, the auxiliary field QMC code of the ALF project allows to simulate a large class of non-trivial models, both efficiently and at minimal programming cost. There are many

possible extensions which deserve to be considered in future releases. The model Hamiltonians we have presented so far are imaginary-time independent. This however can be easily generalized to imaginary-time dependent model Hamiltonians thus allowing, for example, to access entanglement properties of interacting fermionic systems [33–35, 93]. Generalizations to include global moves are equally desirable. This is a prerequisite to play with recent ideas of self-learning algorithms [94] so as to possibly avoid the issue of critical slowing down. Parallel tempering schemes are equally desirable, so as to possibly alleviate long autocorrelation times. Most of the above has already been tested and will be available in the next major release of the ALF package.

On the longer term, we foresee further possible developments. At present, the QMC code of this package is restricted to discrete HS fields such that implementations of the long-range Coulomb repulsion – as introduced in [28, 65, 66] – are not yet included. Extensions to continuous HS fields are certainly possible, but require an efficient upgrading scheme such as hybrid molecular dynamics [45]. An implementation of the ground state projective QMC method within ALF is equally desirable. Efforts in the above directions will be pursued on the longer term.

As it stands, programming a new model certainly requires some detailed knowledge of the algorithm. To facilitate access we hope to maintain an increasing number of model Hamiltonians in the ALF repository. A further step is to aim at cross compatibility with other major projects, especially the ALPS [58] project.

# Acknowledgments

We are very grateful to S. Beyl, M. Hohenadler, F. Parisen Toldin, M. Raczkowski, T. Sato, J. Schwab, Z. Wang, and M. Weber for constant support during the development of this project. We equally thank G. Hager, M. Wittmann, and G. Wellein for useful discussions and support. FFA would also like to thank T. Lang and Z. Y. Meng for developments of the auxiliary field code as well as T. Grover for many discussions.

**Funding information** MB thanks the Bavarian Competence Network for Technical and Scientific High Performance Computing (KONWIHR) for financial support. FG and JH thank the SFB-1170 for financial support under projects Z03 and C01. FFA thanks the DFG-funded FOR1807 and FOR1346 for partial financial support. Part of the optimization of the code was carried out during the Porting and Tuning Workshop 2016 offered by the Forschungszentrum Jülich. Calculations to extensively test this package were carried out both on SuperMUC at the Leibniz Supercomputing Centre and on JURECA [95] at the Jülich Supercomputing Centre. We thank both institutions for generous allocation of computing time.

# A   License

The ALF code is provided as an open source software such that it is available to all and we hope that it will be useful. If you benefit from this code we ask that you acknowledge the ALF collaboration as mentioned on our homepage alf.physik.uni-wuerzburg.de. The Git repository at alf.physik.uni-wuerzburg.de gives us the tools to create a small but vibrant community around the code and provides a suitable entry point for future contributors and future developments. The homepage is also the place where the original source files can be found. With the coming public release it was necessary to add copyright headers to our source files. The Creative Commons licenses are a good way to share our documentation and it is also well accepted by

publishers. Therefore this documentation is licensed to you under a CC-BY-SA license. This means you can share it and redistribute it as long as you cite the original source and license your changes under the same license. The details are in the file license.CCBYSA that you should have received with this documentation. The source code itself is licensed under a GPL license to keep the source as well as any future work in the community. To express our desire for a proper attribution we decided to make this a visible part of the license. To that end we have exercised the rights of section 7 of GPL version 3 and have amended the license terms with an additional paragraph that expresses our wish that if an author has benefitted from this code that he/she should consider giving back a citation as specified on alf.physik.uni-wuerzburg.de. This is not something that is meant to restrict your freedom of use, but something that we strongly expect to be good scientific conduct. The original GPL license can be found in the file license.GPL and the additional terms can be found in license.additional. In favour to our users, the ALF code contains part of the lapack implementation version 3.6.1 from http://www.netlib.org/lapack. Lapack is licensed under the modified BSD license whose full text can be found in license.BSD. With that being said, we hope that the ALF code will prove to you to be a suitable and high-performance tool that enables you to perform quantum Monte Carlo studies of solid state models of unprecedented complexity.

The ALF project's contributors.

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
