# Peer review of "The ALF (Algorithms for Lattice Fermions) project release 1.0. Documentation for the auxiliary field quantum Monte Carlo code"

_SciPost Physics, doi:SciPost Phys. 3, 013 (2017)_

## Round 1 · Referee Report · Anonymous · 2017-5-8

Strengths

1) Highly useful tool, extremely valuable and helpful and welcome
2) Very versatile, large class of models
3) Very impressive complete package, documentation, and examples.
4) Flexible re-binning tool for error analysis.

Weaknesses

Minor.
1) Introduction could be more clear on which models can and cannot be simulated with the present software
2) Autocorrelations: more guidance needed in a black box package
3) Information sometimes a bit terse

Report

This software package for auxiliary field quantum Monte Carlo is an extremely valuable and helpful and welcome tool which had been sorely missing from the existing code packages. It is a very impressive complete package with software, documentation and examples, ready to simulate a very large class of Hamiltonians. This referee would like to thank the authors for providing such a big service to the community.

The documentation is clearly written and complete. Some small additions would be welcome in order to improve accessibility and ease of use of the package, as explained below:

The introduction first provides a quick but very wide review of existing techniques and some corresponding models. The switch to the present package is a bit fuzzy and may leave the reader wondering which aspects of the previous review are included. The subsequent definition of the Hamiltonian is technically complete, but it would be really helpful to have a brief discussion of some models which are or are not included.
In the example section, some repeated information would help, escpecially about which files contain which kind of information (input, output).
The present package can be used almost as a black box. Therefore, some guidance is necessary to prevent accidents. Guidance is nicely provided with respect to stabilization, but not enough with respect to autocorrelations (see below).

Overall, with a few additions in the documentation, I highly recommend publication of this paper.

Requested changes

1) Chapter 1.2: mention some examples of which models can or cannot be simulated. Mention that "flavor" includes spin. Might mention that H_I formally includes the Hamiltonians of H_T.
2) Example section: could repeat which files contain what kind of information.
3) Autocorrelations: Need more discussion. There is no unique autocorrelation time T_auto. The autocorrelation time for convergence depends on the observable and can be very different for e.g. energies vs. long distance correlations. Thermalization usually needs much more than a single T_auto. Like expressed in the paper for the case of stabilization, the user needs to know that convergence ,,has to be judged for every simulation'', and indeed every observable (see for example the ALPS package). The possibility of re-binning is very nice. Some hint would be helpful on how fast to increase bin-sizes and on how many independent bins are necessary for a reliable jackknife error.
4) Page 4: Re(sign) instead of |sign| ?
5) Jackknife: might mention how the sign is handled (eq. 24).
6) For stabilization: scaling with L and beta, timing ?

  • validity: top
  • significance: top
  • originality: high
  • clarity: high
  • formatting: perfect
  • grammar: excellent

Author:  Martin Bercx  on 2017-06-26  [id 151]

(in reply to Report 1 on 2017-05-08)

We would like to thank the referee for the very positive evaluation and their constructive report.
In the following we respond to the points raised by the referee:

1)
"Chapter 1.2: mention some examples of which models can or cannot be simulated."
At the end of Section 1.2, we have added a paragraph mentioning a class of models that cannot be simulated with ALF. The set of shortcomings listed in this paragraph will actually define future directions for the development of our package.

"Mention that "flavor" includes spin."
We have included the following paragraph (Section 1.2):
"Both the color and the flavor index can describe the spin degree of freedom, the choice depending on the spin symmetry
of the simulated model and the HS transformation. This point is illustrated in the examples, see Sec. 4.1 and 4.2."

"Might mention that H_I formally includes the Hamiltonians of H_T."
Please note that the fermionic hopping term H_T and the term H_I describing fermionic hopping coupled to Ising spins are in general independent terms.

2)
"Example section: could repeat which files contain what kind of information."
We have added the following paragraph to the example section:
"The input files are 'parameters' and 'seeds' (see Tab. 8). The output files are 'info', 'confout', and files with suffixes '_scal', '_eq', and '_tau' that
contain the raw measurements (see Tab. 9)."

3)
"Autocorrelations: Need more discussion. There is no unique autocorrelation time T_auto. The autocorrelation time for convergence depends on the observable and can be very different for e.g. energies vs. long distance correlations. Thermalization usually needs much more than a single T_auto. Like expressed in the paper for the case of stabilization, the user needs to know that convergence ,,has to be judged for every simulation'', and indeed every observable (see for example the ALPS package). The possibility of re-binning is very nice. Some hint would be helpful on how fast to increase bin-sizes and on how many independent bins are necessary for a reliable jackknife error"
We agree with the referee that there is no unique autocorrelation time. In the revised version of the manuscript, we have included a much longer version of the discussion of how to determine error bars (see Sec. 2.4). We have also included a highly non-trivial example such that the user can appreciate the intricacies involved in determining autocorrelation times and reliable error bars.

4)
"Page 4: Re(sign) instead of |sign| ?"
Please note that we formulate the weight as |exp[-S(C)]| to account for the, in general, complex phase.

5)
"Jackknife: might mention how the sign is handled (eq. 24)."
The sign is properly handled with the Jackknife method. In the revised version of the manuscript, we have included a paragraph on the Jackknife resampling method (Sec. 2.4.1).

6)
"For stabilization: scaling with L and beta, timing ?"
We have added the following paragraph to the end of Sec. 2.3, which discusses the stabilization:
"While this might seem like quite an effort that has to be performed for every multiplication it has to be noted that even with this stabilization scheme the algorithm preserves the time complexity class of $\mathcal{O}(\beta N^3)$ expressed in the physical parameters inverse temperature $\beta$ and lattice size $N$.
While there is no analytical expression for the dependence of the stability on the physical parameters our experience has been that for a given number of stabilization steps along the imaginary time axis [in the notation of Eq. (36) this number is $L_{\text{Trotter}}/\texttt{NWrap}$], the precision will be largely invariant of the system size $N$, whereas with increasing inverse temperature $\beta$ the number of stabilization steps often has to be increased to maintain a given precision.

---

## Round 1 · Referee Report · Anonymous · 2017-5-22

Strengths

1) first open-source code for lattice fermion models
2) impressively versatile framework
3) broad applicability in terms of model Hamiltonians
4) state-of-the-art implementation and documentation

Weaknesses

1) no compatibility with other similar open-source packages likes ALPS, iTensor
2) no roadmap to include obvious extensions like projective zero-temperature QMC approach
3) no community building effort

Report

The manuscript at hand accompanies an open-source project devoted to a state-of-the-art, parallelized implementation of auxiliary field quantum Monte Carlo approaches for lattice fermion models. Its primary purpose is to provide a documentation of this Fortran 90 code package and illustrate via a number of examples its impressive versatility that allows for the simulation of a broad variety of fermionic lattice models.

This is an important contribution that complements related open-source projects such as the ALPS project or the iTensor package that have been widely utilized over the last decade and found broad appreciation beyond the numerical community. Nevertheless, there is no standardized approach to credit the original authors of such open-source codes in the scientific community and it is therefore of paramount importance to allow for the publication of an accompanying documentation such that scientific credit can be measured in terms of citations.

I therefore strongly endorse the publication of the manuscript at hand in its current form with some minor revisions (see below).

Requested changes

While the manuscript is basically a documentation accompanying an open-source code, it is generally very well written. But I do want to urge the authors to expand the introduction to (i) include a broader introduction into the topic of open-source codes and community codes, and (ii) make a connection to similar efforts such as the ALPS project or the iTensor project (and probably motivate why there is no cross-compatibility with these codes).

One minor point of revision regards the discussion of the sign problem. In particular, Equation (27) should not be credited to the Troyer&Wiese PRL (as done in the introduction), but should be credited to the Loh,Gubernatis, … PRB of 1990.

When referring to introductory readings on fermionic QMC flavors, I suggest to add the very nice book of Gubernatis, Kawashima and Werner (Cambridge University Press, 2016).

In the discussion at the end of the manuscript, I would have expected to find a roadmap for future releases and whether the ALF project is expected to turn into a community project (similar to the ALPS project) with contributors beyond the Würzburg group.

  • validity: high
  • significance: high
  • originality: high
  • clarity: good
  • formatting: good
  • grammar: excellent

Author:  Martin Bercx  on 2017-06-26  [id 150]

(in reply to Report 2 on 2017-05-22)

We would like to thank the referee for carefully reading our manuscript and for the very positive report.
In the following we respond to the points raised by the referee:

"i) include a broader introduction into the topic of open-source codes and community codes, and (ii) make a connection to similar efforts such as the ALPS project or the iTensor project (and probably motivate why there is no cross-compatibility with these codes)."
In the revised version of the manuscript, we have added in the introduction a paragraph placing ALF in the broader context of open source packages for correlated matter. In the conclusions we have touched upon issues of compatibility with other packages. This is certainly desirable, but will be left for future developments.

"One minor point of revision regards the discussion of the sign problem. In particular, Equation (27) should not be credited to the Troyer&Wiese PRL (as done in the introduction), but should be credited to the Loh,Gubernatis, … PRB of 1990."
We have included the above reference.

"When referring to introductory readings on fermionic QMC flavors, I suggest to add the very nice book of Gubernatis, Kawashima and Werner (Cambridge University Press, 2016)."
We have added the reference.

"In the discussion at the end of the manuscript, I would have expected to find a roadmap for future releases and whether the ALF project is expected to turn into a community project (similar to the ALPS project) with contributors beyond the Würzburg group."
At the end of the manuscript, we have added a road map for future releases and new features we plan to add to the ALF package. In the introduction, we have mentioned the efforts we are following to establish a community beyond the Würzburg group.

---

## Round 2 · Referee Report · Anonymous · 2017-7-14

Strengths

Great paper. See first report.

Weaknesses

Earlier minor weaknesses taken care of in the revised version.
Eq (23) uses the real part for the sign of a configuration, whereas on page 1 the modulus is specified; this could be clarified.

Report

I am happy with the revised version of this excellent paper and strongly recommend publication.

Requested changes

See above.

  • validity: top
  • significance: top
  • originality: high
  • clarity: top
  • formatting: perfect
  • grammar: excellent

Author:  Martin Bercx  on 2017-07-17  [id 154]

(in reply to Report 1 on 2017-07-14)

We would like to thank the referee for their positive evaluation of the revised version.
In the following we respond to the point raised by the referee:
"Eq (23) uses the real part for the sign of a configuration, whereas on page 1 the modulus is specified; this could be clarified."

While the configuration weight $e^{-S(C)}$ will in general be a complex number we have the freedom to factor out its real part (Eq. 24).
This is motivated by the observation that the partition function can always be written as a sum of real numbers (see also footnote 4).
The implementation then has the advantage of returning average signs that are real (Eq. 25).

---

## Round 2 · Author Response

Dear Editor,
We would like to thank the referees for their pertinent comments, and their appreciation of providing such an open source package to the community. In the second version of the documentation of the ALF package, we have addressed the points raised by the referees, and detailed them in the replies to the referees (see version 1 of the manuscript). We hope that the manuscript is now in a form that can be published in SciPost.
With best regards,

M. Bercx, F. Goth, J.S. Hofmann, and F.F. Assaad.

---

## Round 2 · List of Changes

List of major changes:
1) Section 1.1: We have included a paragraph on similar open source program packages for correlated quantum matter.
2) Section 1.2: We have extended the discussion on the properties of the general Hamiltonian and we have included a description of the scope and the limitations of ALF.
3) Section 2.3: We have extended the discussion on numerical stabilization.
4) Section 2.4: We have largely extended the section on Monte Carlo sampling and autocorrelations, including a detailed example of error estimation.
5) We have added the computation of autocorrelation functions to the error analysis program of ALF.
6) Section 6: We have extended the discussion on future developments of ALF.

---

## Editorial Decision

published